# Methods to Induce Analgesia and Anesthesia in Crustaceans: A Supportive Decision Tool

**DOI:** 10.3390/biology12030387

**Published:** 2023-02-28

**Authors:** Guiomar Rotllant, Pol Llonch, José A. García del Arco, Òscar Chic, Paul Flecknell, Lynne U. Sneddon

**Affiliations:** 1Institut de Ciències del Mar, Spanish National Research Council (CSIC), Passeig Marítim de la Barceloneta, 37, 08003 Barcelona, Spain; jagarcia@icm.csic.es (J.A.G.d.A.); ochic@icm.csic.es (Ò.C.); 2Department of Animal and Food Science, Universitat Autònoma de Barcelona, Travessera dels Turons, s/n, Cerdanyola del Vallès, 08193 Barcelona, Spain; pol.llonch@uab.cat; 3Comparative Biology Centre, The Medical School, Newcastle University, Newcastle Upon Tyne NE2 4HH, Tyne & Wear, UK; p.a.flecknell@ncl.ac.uk; 4Department of Biological & Environmental Sciences, University of Gothenburg, P.O. Box 463, SE-405 30 Gothenburg, Sweden; lynne.sneddon@bioenv.gu.se

**Keywords:** shrimps, prawns, lobsters, crayfishes, crabs, anesthesia, analgesia, pain, welfare

## Abstract

**Simple Summary:**

Animals that are capable of suffering from pain and distress should be protected when they are being submitted to circumstances that could provoke suffering, such as experimental surgery or killing for human consumption. In terrestrial animals used for scientific purposes and food production, evidence of their capacity to experience pain has led to their legal protection. Recent studies have suggested a pain-like experience in decapod crustaceans. As a consequence, the UK Government has recently recognized decapods as sentient beings. Similarly, some countries have imposed recommendations for the handling, transport, and stunning prior to killing of decapods (Australia, New Zealand, Norway, and Switzerland), acknowledging that suffering during slaughter has a high risk for their welfare. Drugs and methods that may act as anesthetics rendering crustaceans unconscious as well as analgesics or pain-relievers are reviewed here, and other non-decapod crustaceans such as amphipods, brachiurids, branchiopods, copepods, ostracods, and isopods are also included. We developed a detailed on-line tool available publicly that scientists and other stakeholders can employ to search for the most effective methods that effectively anesthetize different crustacean species. This novel tool will also help to identify gaps in existing knowledge so that new drugs or species can be explored in future studies.

**Abstract:**

Methods to induce analgesia and anesthesia for research purposes, handling, transport, or stunning have been used in 71 species of crustaceans. A non-systematic literature search was conducted on crustacean anesthetic methods. This review presents a comprehensive evaluation of drugs and non-chemical methods used to provide analgesia and anesthesia in many crustacean species rather than just decapod crustaceans. This information allows users to select an appropriate method or agent for their species of interest. We prepared an on-line tool based on datasette, a no-code open-source solution for simple web-based database frontends that allows exploration and downloading data by method, analgesic/anesthetic, species, life stage, or sex, as well as other data including environmental conditions (temperature, salinity, light), route of administration, dosage, and induction and recovery times. These values can be selected to filter the dataset and export it to CSV or JSON formats. Currently, several techniques and chemicals are, in our opinion, unsuitable for use as anesthetics in crustaceans, and the basis for these opinions are presented. Given the evidence of a pain-like experience in crustaceans, we propose that researchers should treat crustaceans humanely, applying the principles of good handling, care, and the management of stress and pain to safeguard their welfare.

## 1. Introduction

Recent empirical studies have provided evidence consistent with the occurrence of pain in decapod crustaceans such as crabs, lobsters, prawns, shrimps and crayfish (see review by Elwood) [1]. Scientific studies have shown that crabs exhibit pain-like responses when damaged. Crabs injected with formalin, a chemical used to induce pain in vertebrates, into the claw respond to the injury by intense rubbing of the injured claw during the first 3 min after injection [2]. In saline-treated animals, this reaction was not observed. This suggests that the crab is attempting to reduce a painful effect of the stimulus by grooming or rubbing the affected area. This rubbing directed activity may be a pain-related behavior and might indicate an awareness of the pain that is more than a simple reflex withdrawal reaction. These adverse changes in behavior were accompanied by significant changes in activity in neurons and neuropils of the brain and thoracic ganglia of the crabs which were not seen in saline-injected crabs. These results demonstrate that crabs treated with formalin show both changes in their central nervous system and behavioral changes that are consistent with the definition of animal pain [3]. Decapods fulfill 14 of the 17 criteria proposed by Sneddon et al. [3] for an animal to be considered capable of pain perception. The remaining three criteria have yet to be tested. In 2021, the Department of Environment, Food, and Rural Affairs (DEFRA, UK) commissioned a report that concluded decapods were sentient beings with sufficient evidence for pain and suffering in these animals to propose their inclusion into animal welfare legislation [4]. As a consequence, the sentience of all decapod crustaceans is now officially recognized by the UK Government in the Animal Welfare Bill (Bill [HL], 2022) [5]. Currently, decapods are not included in European legislation governing scientific use of animals which is limited to vertebrates and cephalopods. However, some countries have imposed a ban on the practice of boiling lobsters alive and have introduced directives or recommendations for handling, transport, and stunning (Australia, New Zealand, Norway, and Switzerland).

Very few studies have investigated the capacity for pain in non-decapod crustaceans e.g., amphipods, brachiurids, branchiopods, copepods, ostracods, and isopods. A noxious stimulus, electric shock, was applied to the amphipod, *Gammarus fossarum*, and elicited anxiety where the shocked animal increased time spent in shelter [6]. When six repeated shocks were applied, the animals exhibited increased sheltering for 90 min, demonstrating a prolonged change in behavior. Thus, further studies are necessary to determine if non-decapods also show behavioral and physiological responses to noxious stimuli, but lack of evidence does not negate the possibility of pain in these animals. 

Crustaceans are used as experimental models across a wide range of studies that can be invasive and cause tissue damage [7]. It would be prudent to treat these animals in a humane manner, rendering them unconscious or sedated before and during potentially stressful or painful treatment, and if pain is not the objective of the study, then it may be useful to administer pain relief. A recent review by de Souza Valente provides valuable information on how to anesthetize decapods and the drugs and doses that are effective [7]. Here, we review the agents and methods used for all crustaceans, including non-decapods, and additionally list the methods and chemicals that have been used in studies that we consider are inappropriate for use as anesthetics. Drugs that may act as analgesics or pain-relievers are also reviewed. We also provide details of an on-line tool available publicly that scientists and other stakeholders can employ to find the most effective approach to effectively anesthetize their chosen crustacean species where empirical evidence exists.

## 2. Materials and Methods

A non-systematic literature search was carried out by using review articles on crustacean or invertebrate anesthetics and welfare and the references cited in these reviews [4,7,8,9,10,11,12,13,14,15,16,17,18]. The search was performed between January and August 2022. A systematic literature review was not performed since the term “crustacean” is not always included in the title, abstract, and key words, and because grey literature such as reports, books, book chapters, and papers published in local journals, e.g., from China or Brazil, which were included in this review, may not be identified in automatic searches.

We reviewed 240 references including articles, reports, and books that we found in relation to analgesia and anesthesia, but also narcosis, immobilization, paralysis, and sedation. These latter terms seem often to be used to imply that these agents can be used as anesthetics, but some of the agents cited include sedatives, tranquillizers, neuromuscular blockers (NMBs), preservatives, fixatives, and anti-cholinergic or anti-helminthic agents. These agents would not be considered appropriate for use as anesthetics in other animals.

We incorporated all the information into a tool that can be used to identify anesthetics by crustacean species. The tool is based on the Datasette tool (https://datasette.io/; accessed on 2 February 2023), a no-code open-source solution for simple web-based database frontends that allows exploring and publishing data. Datasette provides plugins to convert excel or csv data to the sqlite database format—the needed input file. Sqlite data tables are shown in a web along with facets, a list of tables showing the summary data from references in different columns. These values can be selected to further filter the table. Finally, output filtered data can be exported to CSV or JSON formats.

## 3. Results

### 3.1. Analgesia

Analgesia is the inability to feel pain and is typically mediated by administration of a pain-relieving drug. In crustaceans, analgesia has received little attention, as shown by the small number of entries that could be included in the tool addressing analgesia (*n* = 87) compared to the references addressing anesthesia (*n* = 1867) (see supportive decision tool and Table 1). An explanation of this small number of studies using analgesics is likely that pain in crustaceans is a relatively understudied field [1,3,18,19].

In crustaceans, most of the references that sought to prevent pain in crustaceans (i.e., analgesia) have used opioids. Opioids include both compounds derived from opium, such as morphine, and also other synthetic and semisynthetic compounds such as fentanyl. In animals, opioids act by binding to opioid receptors, which are distributed widely within the central and peripheral nervous system. Activation of opioid receptors causes membrane hyperpolarization through the activation of potassium channels and subsequent deactivation of calcium channels. The reduction in calcium ions causes a decrease in release of neurotransmitters that are essential for the transmission of the signals of identified pain, so opioid receptor activation induces a strong analgesic effect [31].

The only opioid reported in crustaceans was morphine. Morphine was injected at doses ranging from 2.5 to 150 mg g^−1^ (see tool for specific details). Morphine was first used by Maldonado and Miralto [28] to induce analgesia in mantis shrimps (*Squilla mantis*). Later, morphine has been used in decapod species such as *Carcinus aestuari* [21], *Carcinus maenas* [22], *Chasmagnathus granulatus* [23,24,25], and *Orconectes rusticus* [26,27]. Morphine reduced the sensitivity to electric shock. Administration of an opioid antagonist (Naloxone) blocked this effect [21,23,24,28]. In *O. rusticus*, morphine induced long-term behavioral desensitization but was not related to an analgesic effect [26,27]. In fact, Barr and Elwood [22] could not confirm an analgesic effect of morphine since it reduced responses to light irrespective of whether animals received an electric shock or not [22]. Hence, the effect of morphine in crustaceans needs to be studied further to establish its effects in these species.

Tricaine Methane Sulfate (MS 222) inhibits the initiation and propagation of action potentials by blocking voltage-sensitive sodium channels [32]. In the gammarid *Gammarus pulex*, the analgesic-like effect of 600 mg·L^−1^ (MS 222) during 45 min was evidenced in the lack of behavioral response (use of refuges) of gammarids exposed to an electric shock (10 pulses of 12 V and 2 s) under anesthesia compared to unanesthetized ones [20].

Lidocaine, previously known as lignocaine, is a local anesthetic of the amino amide type. Its mechanism of action is based on the alteration of the signal conduction in neurons by inactivating Na+ channels in the neuronal cell membrane responsible for action potential propagation [33]. In prawns, lidocaine has been used as a local anesthetic for pain relief during eyestalk ablation for experimental purposes. The analgesic properties were induced immediately when 2.5% of lidocaine was applied topically to the eyestalk for *Litopenaeus vannamei* [29] or 5% for *Macrobrachium americanum* [30]. After eyestalk ablation, *L. vannamei* [29] showed no behavioral signs of pain, whereas in *M. americanum* [30] there were some responses such as tail flicking, rubbing, non-sheltering, and recoil, which can be consider as pain indicators. These responses were, however, reduced in prawns that received lidocaine before eyestalk ablation.

The smalll number of studies on analgesia indicates the need for further studies, including more compounds other than morphine, MS 222, and lidocaine. This would both increase our understanding of pain in crustaceans and enable better pain control following surgical or other traumatic procedures.

### 3.2. Anesthesia

#### 3.2.1. Chemical Anesthetics

Flecknell [34] defines anesthesia as, “a state of controllable, reversible insensibility in which sensory perception and motor responses are both markedly depressed”. Nevertheless, we included any publication which referred to narcosis, immobilization, paralysis, sedation, and anesthesia in crustaceans, since these terms were often used to imply production of anaesthesia. This provided a list of 82 different chemical products (Tool, Table 2). Early publications related to anesthesia used chemicals that narcotized (stupor or drowsiness) or immobilized crustacean from zooplankton or marine samples to study their morphology and anatomy with the aim of maintaining the animals in a relaxed state [35,36,37,38,39,40,41,42,43,44]. Some of the products that were used cannot be considered to provide anesthesia. However, the studies employing these agents claimed an anesthetic effect, and thus we highlight them in this review so researchers can avoid using them for the purpose of anesthesia. To do this, we have included a section of non-suitable methods and discuss only anesthetics that we consider effective. Researchers can then choose an appropriate anesthetic drug depending upon the species used and the focus of their research.

Chemicals in this section have been classified in 12 groups: drugs acting at adrenoreceptors, alcohols, esters, steroids, inhalant (gaseous), inhalant (volatile), injectable, oils, opioids, other organic compounds, salt, others; and the results obtained are described below. We assume here that the drugs described below have a mode of action similar to that found in vertebrates (primarily mammals), but this may not be the case, and future studies should verify the action of each drug in crustaceans. To produce general anesthesia in mammals, compounds must cross the blood–brain barrier and act on cerebral tissue. Although a blood–brain barrier is present in some crustaceans [113,114], it has a different structure to that in mammals. In addition to differences in the pharmacokinetics of anesthetics in different species, due for example to differences in tissue composition, there will also be major differences in neuronal receptor populations. Nevertheless, it seems reasonable to assume that drugs that have no demonstrable anesthetic actions in mammals—but immobilize them due to blocking neuromuscular transmission—are unlikely to act differently in crustaceans. However, since a number of publications imply that they have an anesthetic effect in crustaceans, they have been included in the review. We have indicated whether the authors of these publications recommended their use as anesthetics, and whether this is consistent with our current understanding of their mechanisms of action in other species.

There are two main categories of anesthesia: general anesthesia that produces loss of consciousness, and local anesthesia that affects only specific regions of the body [34]. Drugs can be administered via injection into the body, inhalational, or via immersion in water, and this is highlighted below to provide information on the route of administration.

Drugs acting at adrenoreceptors

**Chlorpromazine hydrochloride** is a phenothiazine derivative; chemically, it is the hydrochloride of 3 chloro-10 (3-dimethylamino-n-propyl) phenothiazine [115]. Chlorpromazine has been used as a local anesthetic during surgery in the fields of obstetrics and gynecology, pathology, pediatrics, and psychiatry. There appears to be only one report of its use in crustaceans [45]. Administration to two species of crab (*C. pagurus* and *C. maenas*) resulted in excitement and autotomy of the appendages.

**Xylazine** is a nonopioid veterinary anesthetic and sedative [16]. Its use is described in vertebrates and involves the activation of presynaptic α2 receptors and synaptic inhibition on the central nervous system. Xylazine is the only adrenergic receptor tested in crustaceans [45,46]. Xylazine (70 mg·kg^−1^) administered to three species of crabs produced anesthesia in 5–10 min. True crab species (*Cancer pagurus* and *C. maenas*) had a rapid recovery, whereas recovery was prolonged in the hermit crab (*Coenobita clypeatus*), taking 5–6 h. Reversal of xylazine with a specific antagonist (e.g., atipamezole) to speed recovery seems not to have been reported.

Alcohols

Crustaceans can be anesthetized with various alcohol compounds or mixtures, although the most commonly used has been ethanol.

**2-phenoxy ethanol** is a glycol ether used as a perfume fixative, insect repellent, antiseptic, solvent, preservative, and also as an anesthetic in fish aquaculture where induction and recovery is rapid [116]. In crustaceans, it has been used by immersion in prawns, spiny lobsters, and crabs, including both freshwater and marine species [48,49,50,78]. 2-phenoxy ethanol was effective in juveniles lobsters of *Sagmariasus verreauxi* [48], at a concentration of 2% 2-phenoxy ethanol. This produced anesthesia in around 6 min, and the animal recovered over 4 h. 2-phenoxy ethanol may be effective in other species, but this remains to be tested.

**Ethanol** is primarily a central nervous system (CNS) depressant, and in vertebrates can cause immobility in response to a noxious stimulus [117]. Lo Bianco [35] was the first to suggest the use of 70% alcohol to immobilize different crustacean groups (Ostracoda, Copepoda, Cirripedia, Cumacea, Isopoda, Amphipoda, and Schizopoda) to preserve them in collections to enable further studies of their morphology and anatomy. In particular, for the cirripeda of the genus *Lepas* and *Gonehoderma*, Lo Bianco [35] suggested reducing the ethanol concentration to 35% to keep the cirrhi distended. Patin [51] recommended immersing freshwater species in 10% ethanol for durations ranging from a few minutes to one hour to preserve them. For specific groups of crustaceans (Onychopoda, Cladocera, Ostracoda, Copepoda, Isopoda, Amphipoda, and crayfishes), Pennak [41] recommended use of much higher concentrations (50–95%) and the addition of 5% glycerin when preserving copepods.

*Daphnia magna*, a freshwater species, was completely anesthetized (animals lying motionless on the substrate with a clear reduction in heart rate) following exposure to 10% ethanol for 1 min [38]. In the same species, McKenzie [52] found that the concentration of ethanol to induce anesthesia was influenced by the temperature. A high concentration of ethanol (69%) was needed to produce anesthesia at 5 °C, but 16% ethanol was sufficient at 30 °C. Exposure to 6% ethanol produced immobilization of less than 24-hour-old individuals of *D. magna* over a period of 48 h [53]. Whether age is relevant for anesthesia in daphniids needs to be further evaluated.

Ethanol (40% and 60%) produced anesthesia in *Artemia franciscana*, but the time to anesthesia varied among treatment groups, and exposure resulted in an increase in abnormal behavior [54]. Hence, these authors did not recommend the use of ethanol to induce anesthesia in *A. franciscana*. However, in the copepod *Cyclops* sp., a concentration of 10% ethanol induced anesthesia in 2 min and the copepods recovered in 5 min [55].

Several studies have used ethanol to dissolve water-insoluble eugenol and essential oils before administering to crustaceans [56,57,58,59,60,62,63]. Separate trials were included to demonstrate that the highest-working concentration of ethanol alone did not induce anesthesia and had no observable effect compared to the control groups.

**Ether** is a general anesthetic in vertebrates and was widely used for this purpose. Its effects on the central nervous system have been extensively studied [118]. In the freshwater crustacean species tested, low ether concentrations (0.5–2%) induced anesthesia, although the induction time varied: 1 min in *D. magna* [119], up to 30 min in *Eriocheir sinensis* [49], and 110 min in *Cherax destructor* [65], possibly due to slower diffusion times in larger animals. In *A. franciscana*, a low concentration of ether (1.14%) was used in post-larvae, but to reach full anesthesia of the whole population, immersion needed to be as long as 10 h [64]. In a later study, Foley [66] examined the effect of several anesthetics in *Homarus americanus* and rejected the use of ether since either very high concentrations were needed or all doses were ineffective. Whether ether can be used as an effective anesthetic for marine species should be confirmed in future experiments in different species.

**Isobutanol**, or isobutyl alcohol, was shown to have an anesthetic effect in six marine crustacean species [66,67,68,69] but was either lethal [65] or ineffective [49] in the two freshwater species evaluated. The effect of isobutanol was assessed following immersion (dissolved in water) or after injection. Using the immersion technique, Foley [66] recommended the use of 0.15–0.7% isobutanol in *H. americanus*, which produced full anesthesia in 2–11 min with recovery in 28–91 min. In the spiny lobster *Panulirus* sp., a concentration of 0.01% isobutanol was determined as the lowest dose that induced rapid (25 min) immobilization at 16.5 °C and 0.05% at 28 °C prior to live lobster transportation [68]. It is possible that *Palinurus* sp. immobilization did not correspond to full anesthesia since the aim of the study was to improve transport and not induce deep anesthesia. In *H. americanus*, isobutanol was also injected to produce anesthesia and Gilgan and Burns [67] assessed the effect of temperature. These authors recommended the injection of 1 μL 10 mg^−1^ for each individual at 15 °C. This produced anesthesia within 120 s with a recovery time of 300 s. At lower temperatures (6–8.5 °C), higher concentrations of 2 and 4 μL 10 mg^−1^ per individual were needed, with induction times of 120 and 60 s and recovery times of 1140 and 11,040 s, respectively. The effect of the temperature on anesthesia using isobutanol and similar concentrations was studied in the stomatopod *S. mantis* [69]. These authors observed differences when animals were caught in different seasons; for instance, in individuals captured in winter and acclimated at 16 °C, the induction time was 213 s with a recovery time of 5280 s, while if they were captured in autumn and acclimated to 17 °C, the induction time was 82 s and the recovery time 7710 s.

**Menthol** derived from peppermint, *Mentha arvensis*, has been used as a local anesthetic for vertebrates and shown to act on the transient receptor potential of melastatin-8 (TRPM8) channels [120]. Early studies aiming to preserve barnacles added crystals of menthol to sea water for several hours to keep the animal in an extended position before fixation [42,51,70]. In freshwater daphniids, Cladocera, or copepods, adding crystals of menthol to their culture media for 1 h had either no effect or caused unfavorable movements before the anesthetic took effect [44]. In the marine crab, *Neohelice granulata*, immersion in concentrations of up to 10 mg·L^−1^ of menthol had no effect [73], whereas in the freshwater crayfish, *C. destructor*, menthol was lethal [65]. In the freshwater prawn *Macrobrachium* spp., time of induction to anesthesia was prolonged and therefore was not recommended by authors [60,71]. However, in the freshwater shrimp *Palaemonetes sinensis*, menthol was effective in the three size classes tested and in a range of temperatures from 8 to 28 °C [108]. These authors recommended concentrations of 0.3–0.5 mg·L^−1^ and highlighted that induction time decreased linearly with increasing water temperature and concentration of menthol, and increased with body weight, whereas recovery times lengthened with concentration and temperature, and became shorter with body weight. Specific data on experimental conditions, induction, and recovery time can be consulted in the tool.

**Methyl alcohol** has been shown to be effective in zooplanktonic crustaceans at 0.1 mL·L^−1^ [44]. In *Daphnia pulex*, the induction time was 1 min, while for copepod species, methyl alcohol was also effective but acted very slowly. In all tested species, temperature had little effect on the time required for methyl alcohol to induce a change in the locomotor activity of the organisms.

**Methyl pentynol** [3-methyl-1-pentyn-3-ol] is a liquid with a noxious odour and a burning taste, with a sedative effect that varies in effectiveness with size and species of fish, as well as with water temperature [15]. The anesthetic effect of methyl pentynol was demonstrated in *H. americanus* and *D. pulex*, but was not effective for freshwater copepods [44,66]. Increasing the dose of methyl pentynol from 3 to 12 mL·L^−1^ reduced the induction time from 4 to 15.8 min and prolonged the recovery time from 28.2 to 71.5 min in the American lobster [121]. In water fleas, raising them from 11 to 16 °C increased the induction time from 1.5–6.5 to 10 min [44].

**Tert-amyl alcohol**, in vertebrates, is primarily a positive allosteric modulator for γ-aminobutyric acid (GABA) A receptors in the same way as ethanol [122]. In crustaceans, it has only been tested in *D. pulex.* In this species, it induces anesthesia but with relatively long induction times (3 to 12 min) that increased as the temperature rose from 11 to 21 °C. Large variations in induction times were observed at the lower temperature [44].

Esters

**Benzocaine** is commonly used as an anesthetic for fish, where reflex reactions to a tail pinch and responsiveness to handling are reduced following immersion [123]. The mode of action of benzocaine is to prevent transient increases in sodium ion permeability, thereby blocking the impulse conduction of nervous tissues [124]. General anesthesia was not induced in decapods by immersion in benzocaine [45,65] except in the crab *Pseudocarcinus gigas*, where 0.08 and 0.24 g·L^−1^ of benzocaine produced paralysis in 120 and 45 min, respectively, followed by rapid recovery (10 min). However, this technique was not recommended since it resulted in autotomy of the crab legs during recovery [50]. When benzocaine was used as a local anesthetic on the antenna of the shrimp *Palaemon elegans* [74], grooming and rubbing of the antenna responses were inhibited by benzocaine after application of chemical and physical noxious stimuli. However, benzocaine did not alter general swimming activity. Thus, topical application can be considered as providing pain relief and local anesthesia to specific areas by blocking nerve transmission.

**Pantocaine** is a tetracaine, also known as amethocaine, which is also an ester. Its anesthetic effect is achieved because it is an allosteric blocker of calcium channel function. At low concentrations, tetracaine causes an initial inhibition of spontaneous calcium release events, while at high concentrations, tetracaine blocks release completely [125]. A dose of 0.05% was used by Gliwicz [43] to anesthetize different types of water fleas including *Eubosmina coregoni*, *Chydorus sphaericus*, *Daphnia cucullata*, *Diaphanosoma brachyurum*, *Eudiaptomus graciloides*, and *Mesocyclops leuckarti* under the following conditions: 16 °C, 2500 lux light intensity.

The **phenylurethane** derivative, hydrochloride of piperidinopropanediol di-phenylurethane or diothane, was used as a local anesthetic. Injection of a 0.25% solution into rabbit cornea or to a human forearm had a local anesthetic effect [126]. Naumann [38] proposed phenylurethane as a suitable anesthetic in *D. magna* with a concentration of 0.025% and 0.125% with an induction time of 15 and 1 min, respectively.

**Procaine** (Novocain(e)), is a local anesthetic in mammals and has also been shown to prevent the generation and conduction of nerve impulse in insects [127]. The effects of procaine were assessed in the crustacean *D. magna* [38], and anesthesia was produced after 1 min with immersion in 10% procaine; nevertheless, it was not recommended because of its marked effects on cardiac function. However, in two species of crabs (*C. pagurus* and *C. maenas*), injection of procaine (25 to 60 mg·kg^−1^) was reported to produce anesthesia with a very short induction time (20 to 30 s), long duration (2 to 3 h), and with slow recoveries (2 to 10 h) [45].

**Tetracaine hydrochloride** (THCl), another local anesthetic, appears to have been assessed only in *D. magna*, producing total immobility after 2 h exposure and causing a dose-dependent reduction in heart rate [75]. At a concentration of 182.94 mg·L^−1^, THCl immobilized 50% of the animals.

**Tricaine Methane Sulfate** (MS 222), which is widely used as an anesthetic in fish, has been tested in 21 species of crustaceans with varying results. Concentrations between 0.5 to 2000 mg·L^−1^ produced anesthesia in *A. franciscana*, *Diaptomus* spp., *Limnocalanus macrurus*, *Diacyclops bicuspidatus*, *Eucypris virens*, *Corophium volutator*, *Echinogammarus obtusatus*, *G. pulex*, *Crangon septemspinosa*, *Hemigrapsus nudus*, *Petrolisthes cinctipes*, and *Pugettia producta* with an induction time of 30–90 min [20,44,45,54,76,81,82,128]. The recovery time ranged from 9 to 90 min, determined primarily by the anesthetic bath concentration rather than the duration of exposure. MS 222 had no anesthetic effects in *D. pulex* and other decapod species, irrespective of the concentration and induction times [44,45,47,49,50,63,66,79,80]. The effectiveness of MS 222 as an anesthetic does not seem to be correlated either to taxonomy or habitat. Even within the same taxonomic group, the concentration of MS 222 to induce anesthesia varied greatly among species; for instance, in the gammarid *E. obtusatus*, anesthesia was produced by 0.5 mg·L^−1^ [77], while in the closely-related species *G. pulex*, 600 mg·L^−1^ was required [20]. MS 222 is acidic and should be buffered with a similar concentration of sodium bicarbonate.

**Urethane** or ethyl carbamate is used as an anesthetic for vertebrates because of its minimal effects on cardiovascular and respiratory systems and maintenance of spinal reflexes [129]. Urethane potentiates the functions of neuronal nicotinic acetylcholine, gamma-aminobutyric acid(A), and glycine receptors, and inhibits N-methyl-D-aspartate and alpha-amino-3-hydroxy-5-methyl-4-isoxazole propionic acid receptors in a concentration-dependent manner. In an early study [51], urethane (0.8–1.0%) was suggested as a means to preserve animals, including crustaceans. In daphniids, 0.6–1% urethane induced anesthesia within 15 min [37,40], but lower concentrations were ineffective [38,44]. Similar bath concentrations of urethane were effective for copepods and isopods but ineffective for the crayfish *Astacus fluviales* [130]. Administration of urethane by injection or by gavage was an effective anesthetic in *A. fluviales* [130].

It is important to note that urethane is carcinogenic in mammals, and if it is to be used, appropriate precautions must be taken to protect the operator. The carcinogenetic potential in crustaceans has not been established.

Steroids

**Alfaxalone**—also known as alphaxalone or alphaxolone—a progesterone analog, is a synthetic neuroactive steroid that exerts its action by binding to GABA receptors on the neuronal cell surface, affecting cell membrane chloride ion transport [131]. Concentrations between 15 and 100 mg·kg^−1^ of injected alfaxalone were effective in inducing anesthesia in three crabs species, although half the amount was needed in *Callinectes sapidus* [83] compared with *C. pagurus* and *C. maenas* [45].

Inhalant (Gaseous)

In vertebrates, all inhalant general anesthetics alter consciousness, memory, and pain by acting on the CNS and cause a mild to moderate decrease in the cerebral metabolic requirement for oxygen [16]. Inhalant anesthetics can be classified as either gaseous (in this section) or volatile (next section). In aquatic crustacean species, gaseous anesthetics are usually introduced into the holding water.

The first use of **carbon dioxide** (CO_2_) to anesthetize animals was proposed by Patin [51] as follows: “Squirt a little soda water to a siphon into the water in which the animals are living”. The soda water was then used with daphniids and copedods with a CO_2_ concentration of 100 mL·L^−1^ producing anesthesia in few minutes [44]. Higher concentrations (450 mL·L^−1^) and longer times of induction of anesthesia were required in cirripeds and caridean shrimps, possibly due to the larger size of the animals [84]. Smaldon also noted that the induction time for *Balanus* spp. was 70% higher, probably because the former closed their tergital scutal plates when the soda water was added [84]. When CO_2_ was bubbled into a bath at saturated concentrations, crabs and crayfishes were either not anesthetized at 100%, did not recover from anesthesia, or died [50,84,86]; however, CO_2_ had an anesthetic effect on other species of lobsters, crayfishes, and crabs. In Astacideans (*Homarus* spp. and *Astacus* spp.), anesthesia (defined as absence of recorded stimulus-related responses) was observed after 45 min exposure and animals recovered overnight [85]. In the crab *P. sanguinolentus*, using low doses of CO_2_ (165 mg·L^−1^), only 3.8 min were necessary to induce anesthesia with a short recovery time of 1.7 min [87]. Premarathna [87] also suggested that overdoses of CO_2_ can be used effectively for crab euthanasia. Carbon dioxide forms carbonic acid in water, including on the surface of mucus membranes, and this has been recognized as a welfare problem in other species. The peripheral tissue acidosis caused by carbon dioxide excites nociceptors in fish and is likely to be painful [132]. Whether water acidification has the same effect on crustaceans is not known and further studies should be performed to determine whether CO_2_ can be considered a suitable method to anesthetize crustaceans. From a precautionary perspective, acknowledging the limited data available so far, CO_2_ should be considered with care as an anesthetic for crustaceans.

**Nitrogen** was bubbled into a water bath to investigate its effect on the freshwater shrimp *Macrobrachiurn rosenbergii*, but nitrogen had negligible effect as an anesthetic in this species after 15 min [88]. Roth and Øines [86] used deliquefied nitrogen gas for 3.2 min at −60 °C to super-chill crabs; although freezing prohibited the expression of behavior, it also caused irreversible damage to the appendages, and therefore the method was not recommended. However, nitrogen is an inert gas and pumping it into the water could be effective to induce unconsciousness. In other aquatic species, such as rainbow trout [133], studies suggest that for animals immersed in nitrogen saturated water, unconsciousness is induced without showing aversive indicators. The use of nitrogen might be considered a suitable method, but further studies are needed to confirm this.

**Cycloprane** (C_3_H_6_) is a hydrocarbon ring that was discovered in 1882 and it was employed as an anesthetic in 1933. It is an inhalation anesthetic deemed safe, pleasant, powerful, and controllable, which can be used in humans [134]. However, it forms explosive mixtures with air or oxygen and safety concerns resulted in its use being discontinued. In crustaceans, this drug has only been used to anesthetize *A. franciscana* larvae [64]. In this study, 50% of the larvae population were anesthetized at 29.2% atmospheric concentration for 10 h and recovered successfully.

Inhalant (Volatile)

Volatile anesthetics are liquids at room temperature and normal atmospheric pressure. They can be added directly into a water bath to induce anesthesia in most crustaceans. Four volatile anesthetic agents have been evaluated in crustaceans: chloroform, enflurane, halothane, and isoflurane.

**Chloroform** was one of the first general anesthetics used in people, but its use was discontinued because of cardiac and other toxicity [135]. The effects of chloroform in crustaceans appears to be related to the size of the animal being anesthetized. An early study [64] assessed the effects of several volatile anesthetics for transporting large quantities of *A. franciscana* (100,000 to 150,000 larvae in 4 L). This study reported that chloroform was by far the most potent anesthetic tested in comparison with halothane, ethyl ether, n-pentane, or cyclopropane. The dose of chloroform necessary to produce anesthesia in 50 percent of animals (AD_50_) was 0.0682 g·L^−1^ at 10 h. At this concentration, the change in the percentage of larvae anesthetized was negligible after 5 h. Chloroform has been shown to be a more effective narcotizing agent in cladocerans than copepods [44] and that temperature had little effect on the time required for chloroform to induce a change in the locomotor activity of these organisms. Chloroform has been used as an anesthetic in crayfishes. Kleinholz [136] induced anesthesia in *Pacifastacus leniusculus trowbridgii* to enable removal of the sinus gland with high survival after the procedure. In contrast, although chloroform produced anesthesia in *C. destructor* [65], the mortality rate was high (22–14%). In the freshwater crab, *E. sinensis*, chloroform at 1.5 mL·L^−1^ dose produced anesthesia of all crabs within 30 min [49]. In marine decapods, chloroform (1.5–2.5 mL·L^−1^) immobilized *P. gigas*, but with a slow onset (60 min) and prolonged recovery time (>24 h) [50]. It is important to note that use of chloroform in a water bath will lead to exposure of the operator to a hepatoxic agent.

**Enflurane** was shown to be an effective anesthetic in *D. magna*. Immersion in a concentration of 1.415% enflurane anesthetized 50% of the *Daphnia* at 20 °C within an hour. The concentration used is close to the MAC (minimum alveolar concentration) required in humans and many mammals [90]. In *Daphnia*, the concentration required to produce anesthesia is temperature-dependent, and the concentration of enflurane should be decreased to approximately 0.2% with a water bath temperature of 5 °C [52].

Similar efficiency for anesthesia in *D. magna* was observed using **halothane** (1.006%), and it was also influenced by temperature [52,90]. Halothane at a concentration of 41.5 mg·L^−1^ produced anesthesia of 50% of 100,000–15,000 larvae of *A. franciscana*, but exposure for 5 and 10 h was required [64]. In the shrimp *L. vannamei*, increasing halothane concentration in the bath from 0.5 to 2.5 mg·L^−1^ reduced induction time from 6.5 to 2.5 min but increased recovery time from 11 to 17 min [91]. Halothane was administered as vapour to crayfishes (*Astacus astacus*), and of the range of concentrations assessed (0.01–1.0%), 0.5% was considered optimal [79]. In *C. destructor*, a lower concentration (0.1%) was considered appropriate, with a bath temperature of 10 °C. However, caution should be taken when using enflurane since 17% of the individuals did not recover from anesthesia in this study [65].

**Isoflurane** produces anesthesia in mammals at concentrations ranging from 1.5–5% (34). A similar concentration (1.15%) produced anesthesia in 50% of *D. magna* immersed for one hour at 20 °C [90]. In the freshwater crab, *E. sinensis*, a concentration of 20 mL·L^−1^ of isoflurane produced anesthesia in less than 30 min [49].

Injectable

This section includes compounds that are administered by injection to vertebrates. In crustaceans, this route of administration is often impracticable, and the agents are given by dissolving in a water bath. However, for the convenience of classification of agents, they are grouped together in this section whenever their chemical nature section was not present in this study.

**Bupivacaine hydrochloride monohydrate** is used in humans and animals as a local anesthetic [137]. Concentrations of between 16 and 512 mg·L^−1^ have been assessed in *D. magna* and no immobilizing effects were observed after 2 h exposure [75].

**Chloral hydrate** has been used as a general anesthetic in mammals [138]. The use of chloral hydrate in zooplankton (which included crustaceans) was proposed by Naumann, who also reported its use to anesthetize *D. magna* at a concentration of 10 g·L^−1^. However, the duration of anesthesia was very short (1.5 min) [38]. In other species, chloral hydrate had varying effects, inducing effective anesthesia in *D. cucullata* and copepods species at concentrations of 0.5 g·L^−1^ [43], but only weak anesthetic effects were reported in *D. pulex* and copepods [44]. These weaker effects may have been due to low concentration used in this study (0.0001 mL·L^−1^). There appear to be no reports regarding the use of chloral hydrate in large crustaceans, so its effects are uncertain.

**Etomidate** is a carboxylate imidazole derivative with short-acting hypnotic properties together with anesthetic and amnesic effects. Etomidate is an ultrashort-acting, non-barbiturate hypnotic anesthetic agent [139]. Its effects are produced by acting as a positive allosteric modulator on the γ-aminobutyric acid type A receptor, thus enhancing the effect of the inhibitory neurotransmitter γ-aminobutyric acid (GABA) [140]. GABA is the principal inhibitory neurotransmitter within the CNS. Etomidate (16 and 20 mg·kg^−1^) administered by injection into the pericardial sinus of adult crabs (*C. sapidus*) had no detectable anesthetic effects [92]. Propiscin, a commercial formulation of etomidate (0.2%) which is marketed as an anesthetic for fish, has been used as an anesthetic for freshwater crabs (*E. sinensis*) [49]. Exposure of crabs to Propiscin at a concentration of 20 mL·L^−1^ for 30 min had no anesthetic effects. Since a water-soluble commercial formulation of etomidate is available, further studies in other species of crustaceans may be useful [16].

**Ketamine** induces anesthesia by functional disruption of the CNS through marked CNS stimulation, resulting in catalepsy, immobility, amnesia, and analgesia in vertebrates [16]. Ketamine was used as an anesthetic in three decapod species and the mode of application appears to influence its effectiveness. The response intrathoracic injections of ketamine in the crayfish *Orconectes virilis* was highly variable, while an intramuscular injection of 40 mg·kg^−1^ produced anesthesia with an induction time of 54 s and recovery time of 10 min [80]. Intravascular injection (20 mg·kg^−1^) produced light anesthesia in the crab *C. sapidus* and intracardiac ketamine was fatal in 4 of 5 crabs that were anesthetized using this route [92]. This study also included evaluation of the effects of ketamine:xylazine (20:20 mg·kg^−1^, intravascularly), which produced short periods (5–10 min) of deep anesthesia with a rapid induction time (30–40 s). Ketamine and xylazine were also effective in *P. gigas* [50] when administered intravascularly.

**Pentobarbital** is a barbiturate that produces generalized CNS depression [16]. It is an oxybarbiturate with a slow onset of action and moderately long duration of action in mammals. In crabs, 60 mg·kg^−1^ pentobarbital injection induced anesthesia after 2 min in *C. sapidus*, however, crabs had a very rapid recovery (5 s). A dose of 250 mg·kg^−1^ also induced anesthesia in *C. pagurus* and *C. maenas* (duration to reach anesthesia was not indicated), which lasted up to 90 min [45,92]. In the three crab species, the recovery phase was characterized by ataxia, and therefore this anesthetic was not recommended.

**Propanidid** (C_18_H_27_NO_5_) is an ultra-short-acting phenylacetate that was initially used as a general anesthetic in people but was withdrawn due to anaphylactic reactions. It was first tested as an anesthetic for adult crabs (*C. pagurus* and *C. maenas*) at a dose rate of 100 mg·kg^−1^ [45]. Animals were immobilized within 5 to 6 min and recovered within 1 h. These results are encouraging, but since the agent is no longer available as a commercial formulation, it is unlikely that further studies in a wider range of species will be conducted.

**Propofol** or 2,6-diisopropylphenol is an ultrashort-acting injectable anesthetic agent that is widely used in veterinary anesthesia [16]. Propofol produced rapid onset anesthesia in the crab *C. sapidus* [92], but recovery was associated with tremors limb autotomy. This appears to be the only study published to date evaluating this agent in crustaceans.

**Quinaldine** or 2-methylquinoline has been purported to work through mechanisms similar to those postulated for tricaine, but the exact mechanisms of action are unknown. It has been used to anesthetize fish but produced impaired respiration [141]. Quinaldine has been tested in five crustacean species, but it had variable anesthetic effects [44,66,78]. In *M. rosenbergii*, quinaldine (100, 200 and 300 mg·L^−1^) was considered as it did not produce full immobility in all animals [78].

**Tiletamine–zolazepam**. Tiletamine is a dissociative anesthetic, used in vertebrates (including fish), often used in combination with zolazepam, a benzodiazepine (“Telazol”) [16]. In the crab *C. sapidus*, this combination produced rapid onset (20–60 s) deep anesthesia after intravenous injection (30 mg·kg^−1^ tiletamine–zolazepam), with a short duration of action (5–7 min) [92].

Oils

Essential oils are mixtures of volatile compounds isolated by pressing and distillation from a whole plant or plant part [142]. These compounds are mainly derived from three biosynthetic pathways: the mevalonate pathway leading to sesquiterpenes, the methyl-erythritol pathway leading to mono- and diterpenes, and the shikimic acid pathway en route to phenylpropenes. In crustaceans, the anesthetic effects of oils isolated from *Aloysia triphylla*, *Amyris balsamifera*, *Cymbopogon citratus*, *Lippia alba*, *Ocimum gratissimum*, *Origanum majorana*, *Lavandula officinalis*, *Melaleuca alternifolia*, *Mentha piperita*, *Passiflora incarnata*, *Valeriana officinalis*, and mainly *Eugenia caryophyllus* have been assessed [7]. All of these plant oils induced anesthesia, except for *Passiflora incarnata* and *Valeriana officinalis*. These two oils appear only to have been assessed in juvenile prawns (*Macrobrachium tenellum*) following immersion in a water bath for 2 h with an oil concentration of 300–900 mg·L^−1^ [60].

**Clove (*E. caryophyllus*) oil** is a mixture of compounds, of which the main active ingredients are phenolic eugenol (85–95%) and isoeugenol and methyleugenol (5–15%) [16]. Note: methyleugenol has been found to be carcinogenic [143]. AQUI-S^®^ and AQUI-S^®^ 20E are commercial formulations containing 50% isoeugenol and 10% eugenol, respectively (https://www.aqui-s.com; accessed on 2 February 2023). Clove oil and its active compounds are not water-soluble, and so have most frequently been used after being dissolved in ethanol. We compiled the data from studies that used either **clove oil**, **AQUI-S**, or **eugenol**, since the compound that has the greatest anesthetic effects is eugenol. Two reports showed similar effects of clove oil and AQUI-S [50,78]. Throughout this section, we refer to clove oil and AQUI-S also as eugenol.

Eugenol induced anesthesia in daphniids [95,96], amphipods [20,56], penaeid shrimps [57,99,100,102,103,104,105,106,107], caridean shrimps [60,72,78], claw lobsters [62,110,111], crayfishes [49,65,97,107,109], spiny lobsters [48,94], hermit crabs [98], and crabs [50,63,73,87,107,110], but its anesthetic effects were inconsistent in *A. franciscana* [54]. No interrelationship among the taxonomic groups in relation to concentration or induction and recovery time were shown. Eugenol (0.1–0.2 μL·g^−1^) injected into the pericardial sac of lobsters (*H. americanus*) and crabs (*C. maenas* and *Cancer irroratus*) resulted in anesthesia [110] and 10-fold higher doses were considered suitable for euthanasia of these species. In contrast, doses as high of 5 μL·g^−1^ injected intravascularly were ineffective in inducing in the crab *N. granulata* [73] and also produced autotomy and/or death.

Several factors have been reported to influence the effectiveness of eugenol as an anesthetic in various animal species including crustaceans: concentration of eugenol, animal characteristics (stage of development, size, sex, maturity), and environmental conditions (temperature, salinity). In general, with increasing concentrations of eugenol, the times required for complete anesthesia significantly decreased in all studied crustacean species, while the recovery times increased [50,56,57,59,60,62,63,65,71,72,78,87,96,100,104,105,106,109,111,144]. Eugenol used at identical concentrations resulted in slower induction times in adults of *D. magna* [96], *M. rosenbergii* [59], and *L. vannamei* [57] compared to their earlier stages. In contrast, juveniles of *Penaeus japonicus* had longer induction times than adults [105]. Within the same development stage, induction and recovery time increased with body weight in *Cherax quadricarinatus* [97], *Penaeus monodon* [106], and *Penaeus chiniensis* [72], while no effect on size was reported in *Jasus edwardsii* [94]. The effects of eugenol did not differ between sexes in *J. edwardsii* [94] and in *C. quadricarinatus* [97]. However, it was reported that anesthetic induction was slower and recovery faster in berried (gravid) females of in *N. norvegicus* compared to non-berried [62].

The anesthesia time of the shrimps (*Penaeus semisulcatus*, *P. chiniensis*, *Penaeus monodon*, and *P. japonicus*) decreased with increasing water temperatures, and the recovery time increased with decreasing water temperature [72,100,102,104,106]. Salinity also influenced the effects of eugenol in *P. semisucatus*, and its effects interacted with temperature and eugenol concentration [100].

Other essential oils have been less studied, but some promising results were shown either for light or full anesthesia. ***A. triphylla* (EOAT)** was effective in the shrimp *L. vannamei* [57] but ineffective in the crab *N. granulata* [73]. The concentrations of EOAT recommended for full anesthesia in the shrimp was 0.300 mL·L^−1^ for sub-adults and post-larvae [57]. **Sandalwood *A. balsamifera* (SAN)** induced anesthesia in the shrimp *Penaeus schmitti* in 4.2 min using a concentration of 0.50 mL·L^−1^, but exposure in a bath for 15 min failed to anesthetize the closely related species *Penaeus brasiliensis* [99]. These authors suggested that SAN can be useful as an agent to optimize transport of shrimp. ***C. citratus* (EOC)** was only tested in *L. vannamei*. Concentrations between 0.150 and 0.500 mL·L^−1^ induced anesthesia within 2.5 and 1.8 min with a recovery time of between 4.2 and 6 min [101]. Lower doses between 0.005 and 0.010 mL·L^−1^ were also tested for long-duration anesthesia (6 h), when recovery took between 2 and 4 h, however concentrations above 0.015 mL·L^−1^ produced mortality in 1 h [58]. The concentration of ***L. alba* (EOLA)** recommended for anesthesia was 0.750 mL·L^−1^ for adults of the shrimps *Penaeus pauliensis* [101] and *L. vannamei* [57]. For the post-larvae of *L. vannamei*, the recommended concentration was lower than for adults to induce short anesthesia needing 0.6 mL·L^−1^ of EOLA [57]. These later authors also evaluate the concentrations of EOLA during the transport (6 h) and recommended 0.050 mL·L^−1^ and 0.020–0.025 mL·L^−1^ of EOLA for adults and post-larvae, respectively. Anesthesia was not produced in the crab *N. granulata*, even when using concentrations up to 8 mL·L^−1^ of EOLA [73]. A concentration of 0.10 mL·L^−1^ of ***O. gratissimum* (EOOG)** induced anesthesia in the shrimp *P. pauliensis* with no adverse effects. However, a higher concentration (0.25 mL·L^−1^) of EOOG induced anesthesia but caused muscle spasms, escape responses, and 30% mortality. ***O. majorana* (EOO)** tested in *L. vannamei*, at concentrations of 0.40 and 0.80 mL·L^−1^, induced anesthesia in 2.5 and 1.7 min and showed a recovery time of 5 and 10 min, respectively [101]. Lower doses (0.05 and 0.2 mL·L^−1^) were not effective in the first 30 min. ***L. officinalis* (LAV) and *M. piperita* (MEN)** with a concentration of 0.50 mL·L^−1^ induced anesthesia in *P. brasiliensis* and *P. schmitti* within 2 to 4 min with recovery times of 45 min for *P. brasiliensis* and 28 min for *P. schmitti* [99]. The recovery times for LAV and MEN were longer than those reported for EOAT, SAN, EOC, EOLA, EOOG, and EOO. In the crab *N. granulata*, anesthesia was produced after immersion in ***M. alternifolia* (tea tree oil: TTO)** at a concentration of 8 mL·L^−1^. Anesthetic induction and recovery were relatively slow (20–30 min), and more rapid effects (7 and 9 min) were obtained after injection of nanoencapsulated TTO (4–6 μL·g^−1^) [73]. Hence, these authors suggested that nanoencapsulated TTO might be considered effective as an anesthetic in *N. granulata*.

Other organic compounds

**Acetic paraldehyde (C_6_H_12_O_3_)**. Paraldehyde is the cyclic trimer of acetaldehyde, a colorless or slightly yellow-colored liquid. Paraldehyde has been used historically as a sedative/anesthetic in rabbits [145], and in crustaceans it was tested in four small freshwater species [39]. In the copedods (*Cyclops strenuous* and *Eudiaptomus vulgaris*), 5 min immersion in a concentration of 0.5 g·L^−1^ induced anesthesia after 6 or 8 min, respectively. Longer times and/or a higher concentration of 1 g·L^−1^ was associated with mortality. Daphniids were more resistant to acetic paraldehyde; in particular, exposure of *Daphnia longispina* to 1 g·L^−1^ for 30 min anesthetized them without causing any mortality. A bath with concentration of 2 g·L^−1^ was also effective for *D. longispina* but resulted in high mortality in *Daphnia obtuse*. Ravera [39] recommended it for copepods and cladocerans.

**Chloretone** forms when caustic potash is slowly added to equal weights of chloroform and acetone, and may be isolated from this mixture, after the removal of any excess of acetone and chloroform, by distilling with steam [146]. In crustaceans, chloretone has been used for anesthetizing individuals, keeping them relaxed before fixation. In 1900, Randolph [36] showed that concentrations of chloretone from 0.0002 to 1 g·L^−1^ induced anesthesia in cladocerans, copepoda, and ostracods. For *Daphnia* ssp., he recommended use of 0.5 g·L^−1^ which produced anesthesia in 24 h. Later, Knudsen [70] recommended use of chloretone (0.29 g·L^−1^) in cirripeds for 6 h before fixation. Gannon and Gannon [44] reported that a concentration of 0.00008 mL·L^−1^ of chloretone was ineffective for inducing anesthesia in *D. pulex* and a few copepod species, however, this may have been due to the very low concentration of the agent that was used.

**Clomethiazole** is sold commercially as Hemineurine and is a hypnotic and sedative with anticonvulsant effects [147]. In the crab *C. maenas*, immersion in clomethiazole (2.5 g·L^−1^) induced anesthesia in 5 min with a recovery time of 40 min. However, immersion in a slightly higher concentration (3 g·L^−1^) was lethal, so it must be used with care. In another crab species, *Necora puber*, much lower concentrations—just above 0.01 g·L^−1^—were sufficient to induce anesthesia within 15 min [112].

**Guaiacol glyceryl ether**, also known as Guaifenesin, is a constituent of guaiac resin from the wood of *Guajacum officinale* and has been used as anesthetic in vertebrates [148]. Oswald [45] tested the anesthetic properties of this compound in two species of crabs (*C. pagurus and C. maenas*) by an injection of up to 0.4 mL of a 5% solution. Guaiacol glyceryl did not show any effect on any species of crab.

**Lidocaine** was used to anesthetize water fleas *D. magna* [75], shrimps *L. vannamei* [29,91] and *M. americanum* [30], crayfishes *Orconectes virilis* [80], crabs *C. sapidus* [92], *C. pagurus* [45], *C. maenas* [45], *Portunus sanguinolentus* [87], *P. gigas* [149], and the stomatopod *S. mantis* [69]. Shrimps and crabs have been exposed to many different concentrations of lidocaine by immersion, ranging from 0.4 to 5 g·L^−1^, but lidocaine has also been administered topically (2.5 g·L^−1^) and by intrathoracic injection (0.02 g·L^−1^). Anesthesia was only reported in shrimps (*L. vannamei* and *M. americanum*), which were exposed by water bath and topically. This suggests that intrathoracic injection is not a suitable way of administration. Alternatively, immersion may be suitable, but it seems to be dependent on the species (shrimps and crabs). The concentration was shown to influence the time to reach an effect, with higher concentrations having a more rapid induction effect. In summary, lidocaine administered by immersion has been shown to be an effective anesthetic for shrimps and crabs.

**Piperazine** (C_4_H_10_N_2_) is an organic compound that consists of a six-membered ring containing two nitrogen atoms that is freely soluble in water. The piperazines are a broad class of chemical compounds, which contain a core piperazine functional group. It is used as an anthelminthic for the treatment of nematode infections. It acts at GABA receptors, resulting in muscle paralysis. It was tested by Gliwicz [43] in *E. coregoni*, *C. sphaericus*, *D. ucullate*, *D. brachyurum*, *E. graciloides*, and *M. leuckarti*. Immersion in a 0.05 g·L^−1^ solution resulted in immobilization. Since the mechanism of action of this agent in crustaceans is uncertain, it cannot be recommended as an anesthetic until more information becomes available.

**Suxamethonium chloride** (suxamethonium or succinylcholine) is a short-acting neuromuscular blocking agent that has been used alongside anesthetic agents in people and animals [150]. Its mechanism of action is binding to the nicotinic acetylcholine receptor, resulting in a depolarization of the motor endplate. This occurs when both receptors bind acetylcholine, causing a conformational change in the channel complex, opening it to the inward flow of positive ions and calcium is released. By maintaining the membrane potential above threshold, the muscle cells cannot repolarize. When acetylcholine binds to an already depolarized receptor, it cannot cause further depolarization [151]. It has been used in crabs (*C. pagurus* and *C. maenas*) by Oswald [45]. In that study, adults were injected with an initial concentration of 5 g·L^−1^, rising it to 40 g·L^−1^ of solution of suxamethonium chloride, but no effect was noted.

**Terpinen-4-ol** (C_10_H_18_O) is an isomer of terpineol. It is a primary constituent of tea tree oil (*M. alternifolia*) that is obtained as an extract from their leaves, branches, and bark. Terpinen-4-ol was mentioned by Souza et al. [73] and Patin [51] for its potential use as an anesthetic compounds in crabs (*N. granulata*) and other decapods, respectively. Souza et al. [73] used it in male adult crabs, investigating the effect after administration by injection or by immersion in a water bath. Immersion in a solution of 8 mL·L^−1^ produced anesthesia after 24 min. When administered by injection, concentrations of 0.02, 0.03, and 0.04 mL·L^−1^ induced a full anesthetizing effect, which lasted 39 min, 4 min, and 50 s, respectively. The lowest concentration of 0.01 mL·L^−1^ did not anesthetize these crabs.

Salts

None of the various salts (e.g., magnesium sulphate, sodium chloride) that have been assessed in crustaceans can be recommended for use as anesthetics. Therefore, the information regarding salts is described in the Section 3.2.3.

#### 3.2.2. Physical Methods to Reach Anesthesia (Table 3)

All physical methods reviewed here have been used to produce unconsciousness. Some methods may be reversible, and others produce irreversible loss of consciousness. They may be of interest for researchers and/or producers when chemical methods are not considered appropriate.

Thermic methods

Changes in temperature have been reported to be useful to anesthetize crustaceans either by heating or cooling them in the air or in water. Insensibility occurs more quickly in an ice slurry than in air at similar temperatures because water absorbs heat much faster than air. In this section, we will concentrate on **cooling** methods since we consider that heating induces distress and takes longer to rend the animal unconscious. Further, high temperatures may stimulate nociceptors with temperatures above 29 °C stimulating nociceptors in fishes [181]. Further information on the use of heat is given in the section on non-suitable methods below.

We identified 36 studies (Table 3) that used cooling to reduce mobility and prevent stress, making the animals easier to handle for surgery, transport, and stunning. Cooling is usually undertaken by placing the animals in ice slurry (−1 to 5 °C) or cold water. The temperature of the water used for cooling should be at least 10 °C lower than the rearing/catching water temperature). Cooling was first used to immobilize copepods (*Cyclops* sp.), to allow assessment of the presence and abundance of a parasite [55]. Cooling was also recommended as a useful technique to immobilize crabs when studying their morphology [70].

Cooling has been described for a range of different groups of decapods including penaeid shrimps, caridean prawns, clawed lobsters (Astacidea), crayfishes (Astacidea), spiny lobsters (Palinuridea), hermit crabs (Anomura), and true crabs (Brachyura). The majority of shrimp studies used cooling to improve survival during transport. Live shipment technology was developed and improved in Japan by the 1960s, where wild or cultured Kuruma shrimp, *P. japonicus*, were transported alive [172,173]. This technique has two steps: a pre-chilling step, where animals are placed in water at 12–14 °C, and then shrimps are packed in boxes with pre-chilled sawdust (0 °C or −10 °C) and kept in coolers at 5–10 °C during transport. This two-step cooling technique was also implemented in other shrimp species [88,168,170,171,174,175,176,177,179] and spiny lobsters [12,68]. In a research setting, cooling of shrimps, lobsters, crayfishes, and crabs with ice has been used in an attempt to mitigate pain during surgery or tissue sampling [26,27,85,157,161,162,163,164,166,180,182].

Studies of the use of cooling for transportation showed that its efficiency in producing immobility or anesthesia depends upon several factors. First, the tolerance to cold varies among species even between those that are closely related. For example, the shrimp *P. japonicus* could tolerate low temperature storage at 5 °C [172], while *Penaeus esculentus* required temperature to be maintained above 12 °C [171]. Similarly, the crabs *Portunus pelagicus* cannot safely be cooled below 6 °C [165] while the closely related species *P. sanguinolentus* could tolerate temperatures of 0 °C [87]. Tolerance of cooling also varies with different life stages. Adults of the shrimp *P. monodon* had a survival rate of 98% after use of cooling for transportation, but the mortality was over 20% in juveniles and 10% in larval forms [176]. Cooling rate also had an effect on the survival of shrimps, increasing when the rate was slow in *L. vannamei* [179] and *Penaeus monodon* [183]. Providing supplemental oxygen during cooling and transportation increased survival in *L. vannamei* [179].

The water temperature of the housing environment has been shown to influence the effects of cooling. McRae et al. [65] reported that cooling was effective when the crayfish *C. destructor* was housed at 16 °C, but cooling was less effective when the crayfishes were housed at 10 °C.

Studies assessing the use of cooling to immobilize or anesthetize crustaceans for a range of purposes indicated considerable variation in response in different species. Cooling was considered as effective as an anesthetic agent in *E. sinensis* [49] and *P. sanguinolentus* [87], but was ineffective for *A. astacus* [85], *Astacus leptodactylus* [85], *H. americanus* [85], *Homarus gammarus* [85], *C. pagurus* [86], and *P. gigas* [50].

Anesthesia, assessed by the absence of response to mechanical and electrical stimulation, was not seen after 60 min of cooling lobsters (*Homarus* spp.) and crayfishes (*Astacus* spp.). Cooling caused only a minor reduction in neuronal responses recorded with implanted electrodes [85]. Similarly, in the crab *C. sapidus*, cooling was ineffective since the crabs maintained a sensory–CNS–cardiac or sensory–CNS–skeletal muscle response in cold (<4 °C) for 4 min [152]. Heart rate decreased, quickly when shrimp *L. vannamei* or crayfish *P. clarkii* were transferred to ice, however, for the crab *C. sapidus*, the heart rate decreased slowly. Moreover, shrimp and crayfish showed a sharp decrease in activity within a sensory–CNS–cardiac response when chilled, indicating that they were anesthetized.

Mechanical methods

Methods that mechanically kill crustaceans are usually not considered as analgesics or anesthetics, as they produce an irreversible loss of consciousness. However, mechanical methods have been evaluated as components of slaughter procedures, with the aim of minimizing suffering and pain.

Mechanical methods have been used to kill commercial decapod species. In decapods, the nervous system is composed of 22–23 segmental units or neuromeres, extending from the protocerebrum (partially located in the eyestalks) to the last pleonic neuromere. In adult shrimps/prawns, lobsters, and crayfishes, these neuromeres run down the full length of the body. In crabs, in which the abdomen bends under the cephalothorax during development, neuromeres 4 to 22–23 fuse to constitute the thoracic ganglion [113]. Hence, two mechanical methods to destroy the nervous system have been established, splitting for shrimps/prawns, lobsters, and crayfishes and spiking for crabs [160]. **Splitting** involves rapidly cutting through the centerline of the cephalothorax and abdomen with a large sharp knife, while **spiking** pierces and destroys the two main ganglia. For detailed procedures on splitting and spiking, see The Royal Society for the Prevention of Cruelty to Animals (RSPCA) Australian’s manual. The use of splitting and spiking was previously recommended by the National Aquaculture Council of Australia in those species of crayfish, lobster, and crab used in aquaculture [12]. Spiking has been shown to be effective in crabs for commercial slaughter [86,89]. These latter authors [86] highlighted that spiking must be carried out by trained personnel to assure that the nervous system is completely destroyed.

Electrical methods

Electrical stunning is used as a method of slaughter of crustacean, and a commercial device, “Crustastun” (www.crustastun.com; accessed on 2 February 2023), has been developed for this purpose. Application of the device to deliver a lethal electric shock (110 V, 2–5 A for 5 or 10 s) to lobsters (*N. norvegicus* and *H. gammarus*) and crabs (*C. maenas* and *C. pagurus*) showed that this stunning apparatus silenced the central nervous systems of animals involved [153,157], but this cannot be considered an anesthetic method. There was no measurable additional stress due to the electrical stunning process itself [154] and quality of the meat, in terms of texture and flavor, was improved [158]. To stun crabs (*C. pagurus*), electro shock was applied at 220 V, 50 Hz, 0.65 to 2.2 A for 1 s [86,155,156]. Longer exposure time to the electric shock (30–60 s) increased temperature in the crabs and therefore was not recommended due to burn marks.

Electrical stunning using either “Crustastun” or a second commercially available device (LAVES) was evaluated in the European lobster *H. gammarus* and in two crayfish species, *A. astacus* and *A. leptodactylus* [85]. Both devices were able to stun and render unconscious lobsters and crayfishes. Electrical stunning induced epileptiform seizures followed by a decline in the nervous system activity.

Electrical stunning induced immobility in shrimps (*L. vannamei*), crayfishes (*P. clarkii*), and crabs (*C. sapidus*), which recovered their movements and swimming capacities between 300 and 600 s [152]. However, although heart rates decreased after the electric shock, the shapes of the electrocardiograms (ECG) traces did not regain full recovery to the state they were prior to the shocking, indicating some alteration in the function of the heart.

Lower electric intensities have been used as a noxious stimulus to study the behavior and physiological changes in crustaceans, for example, in the crayfish *P. clarkii* [184,185], the crab *C. maenas* [186], or the amphipod *G. pulex* [20], but these and similar studies are outside the scope of this review.

Electrical stunning can be used to render crustaceans unconscious and issued as a stunning method before a second method is employed to ensure death. The RSPCA in Australia recommends the use of electrical stunning for all crustacean species. The use of Crustastun is ideal for restaurants while large productions will need larger electrical stunners such as those manufactured by Optimar, AceAquatec, or Polar Systems. However, further studies are needed to determine appropriate conditions for the different species of crustaceans.

#### 3.2.3. Non-Suitable Methods (Table 4)

The following methods have been reported to induce anesthesia, but we consider them unsuitable for one of the following reasons: (1) the agents are not anesthetics—for example, they have other functions such as neuromuscular blockers, anti-cholinergic agents, or anthelmintic drugs; (2) they are dangerous for humans and non-human animals and accidental exposure would lead to critical risks for public health; (3) they are too expensive or the quantities required for anesthesia are impracticable; (4) large variability exists in anesthetic effects in different species and conditions.

Chemical anesthetics

Alcohol

**BOUIN** (Alcohol + Formaldehyde + glacial acetic acid + picric acid) has been recommended for the preservation of animals, including crustaceans as Anostraca, Ostracoda, and decapod species [42,51]. It is a histological fixative and not an anesthetic.

Other organic compounds

**Butyn** is a local anesthetic chemically similar to procaine but 10 times stronger, and was used for dentistry in the early 1900s [195]. It has been used in crustaceans (*D. pulex*, *Diaptomus* spp., *L. macrurus*, and *D. bicuspidatus*), but high concentrations were required and the onset of action was too slow for the agent to be of practical use [44].

**Decamethonium bromide** was assessed as an anesthetic for crabs (*C. pagurus* and *C. maenas*), but caused immediate excitement followed by autotomy of appendages [45]. It should also be noted that decamethonium bromide is a neuromuscular blocking agent, not an anesthetic [196].

**Formalin** is a formaldehyde aqueous solution. In crustaceans (Branchiopoda, Amphipoda, Cumacea, Copepoda, Ostracoda, and Malacostraca), as in other animals, it has been used as a tissue fixative [41,42,187]. It does not have anesthetic properties.

**Gallamine triethiodite** was used to immobilize fish during electro-olfactogram procedures to study olfactory receptor neurons [197]. No effect was seen after use of 5 mg·L^−1^ in *C. pagurus and C. maenas* [45]. Since gallamine triethiodite is a neuromuscular blocking agent, it should not be used as an anesthetic.

**Hydrogen peroxide** (H_2_O_2_) is a pale blue liquid, slightly more viscous than water, that is used as an oxidizer, bleaching agent, and antiseptic. As an anesthetic, it was tested by Gannon and Gannon [44] in *D. pulex*, *D. bicuspidatus*, *Diaptomus* spp., and *L. macrurus* in a 3% concentration at 0.01 mL·L^−1^ in a water bath. It showed inconsistent effects: *D. pulex* did not show any effects, whereas *D. bicuspidatus* and *Diaptomus* spp. showed immobility in 5 min, and *L. macrurus* in 2–3 min. It is an irritant compound because of its oxidant capacity and does not have anesthetic properties.

**Hydroxylamina** is a white crystalline, hygroscopic compound with the formula NH_2_OH. Its anesthetic properties were assessed in adult specimens of *D. pulex*, *Diaptomus* spp., *L. macrurus*, and *D. bicuspidatus*, using a concentration of 0.01 mL hydroxilamina in 1 L of salt water [44]. It produced immobility in all species, but it is not clear whether this was due to any anesthetic properties.

**Methysergide maleate** (C_21_H_27_N_3_O_2_), also known as methysergide, is a monoaminergic medication of the ergoline and lysergamide groups that antagonizes the effects of serotonin in blood vessels and gastrointestinal smooth muscle [198]. It has been used for the treatment of migraine in people. Its potential anesthetic effects have been assessed in crayfish (*Procambarus clarkii*) [188]. Animals exhibited anxiety-like behavior and were not completely immobilized. It seems unlikely that this agent has anesthetic effects in any species.

Salts

**Magnesium chloride** (MgCl_2_) was used with adult crayfish (*A. astacus* and *A. leptodactylus*) and American lobster (*H. americanus*), and juveniles of the European lobster (*H. gammarus*) in a study investigating neural response during stunning and killing [85]. For placing electrodes, different anesthetic methods were used, including magnesium chloride, to identify the one most suitable. Exposure to 10% MgCl_2_ for an hour did not sedate any of the species studied. A similar lack of efficacy was reported in *C. destructor* [65]. It seems unlikely that MgCl_2_ can be used as an anesthetic agent for crustaceans.

**Magnesium sulphate** (MgSO_4_) is a salt, soluble in water, and has been assessed as an anesthetic agent in a range of species including water fleas (*D. pulex*), lobsters (*H. americanus*), and crabs (*Hemigrapsus sanguineus*, *P. gigas*, and *P. sanguinolentus*) [2,44,50,66,87]. In all species, animals were placed in a water bath of magnesium sulphate, except for the crab *H. sanguineus*, which was injected. Anesthesia was either not induced successfully, required prolonged (>1 h) exposure, or required excessive concentrations of magnesium sulphate, so this compound is considered not suitable as an anesthetic for crustaceans.

**Physostigmine salicylate** (commercialized as Santonium) is a highly toxic parasympathomimetic alkaloid—specifically, a reversible cholinesterase inhibitor—that paralyses crustaceans as the result of blockade of the synapses of the peripheral neuromuscular system, impeding the synaptic transmission between neurons. It was ineffective in adult *D. pulex* at 80 mL·L^−1^ [44]. Exposure of four species of Branchiopoda including, *E. coregoni*, *D. cucullata*, *C. sphaericus*, and *D. brachyurum*, and one species of copepod *M. leuckarti* produced immobility [43], but since this agent has no anesthetic properties, it should not be used for this purpose.

**Potassium chloride** (KCl) or potassium salt is a metal halide salt. Its use for euthanasia has been reported in two species of decapods (*H. americanus* and *C. pagurus*) [189]. This agent has no anesthetic effects, and in mammals, death is caused by cardiac arrest. It should not be used as an anesthetic and should not be administered to conscious animals.

**Potassium hydroxide** (KOH) is an inorganic compound, commonly called caustic potash, with potent alkaline and corrosive properties. It was used for killing Branchiura spp. by Mahoney [42] with a 10 g·L^−1^ water bath concentration. No time needed to reach an effect was indicated by the author. It has no anesthetic properties, and it should not be used in any animal that may be capable of experiencing pain.

**Sodium bicarbonate** (NaHCO_3_), upon dissociation, forms sodium and bicarbonate ions. Ion formation increases water bicarbonate and buffers excess hydrogen ion concentration, resulting in raised water pH, therefore reducing acidification. It was used in adult *Ranina ranina* [190], but recovery was very prolonged and associated with high mortality. It is not an anesthetic agent in vertebrates and cannot be recommended for anesthesia in crustaceans.

**Sodium chloride** (NaCl) or common salt is the main salt responsible for the salinity of seawater and of the extracellular fluid of many multicellular organisms. The potential anesthetic effects of sodium chloride have been assessed in species of adult decapod crustaceans (*H. gammarus*, *C. pagurus*, and *P. gigas*) [50,86,89,191]. Salt concentrations differed in the four studies, ranging from a saturated solution [89] to 350 g·L^−1^ [191]. An immobilizing effect was reported in *H. gammarus* (within 1 min) and *P. gigas* (within 44 min). No anesthetic effect was noted in *C. pagarus* [86,191], although the animals appeared sedated. Although NaCl has been used to stun and kill aquatic animals, including edible crabs, this method is unsuitable for anesthesia.

Others

**Albumin** is a family of globular proteins that are commonly found in blood plasma [199]. It was reported to have a narcotic or immobilizing effect in adults of *D. magna* [38], but since its effects work by physically limiting the movement of small crustaceans, it should not be considered an anesthetic.

**Cocaine** is a local anesthetic and a stimulant obtained from the leaves of two vegetal species native to South America, *Erythroxylum coca* and *Erythroxylum novogranatense*. After extraction from coca leaves and further processing into cocaine hydrochloride (powdered cocaine), the drug may be dissolved. Cocaine blocks the dopamine transporter, inhibiting reuptake of dopamine from the synaptic cleft into the pre-synaptic axon terminal [200]. Cocaine also blocks the serotonin and norepinephrine transporter and increases activation of serotonin receptors and norepinephrine receptors in the post-synaptic neuron, leading to lack of arousal and euphoria in humans. When applied topically, it has local anesthetic effects in people and animals. It has been used to anesthetize *D. magna* in a water bath [38] with either 0.25, 1 or 10 g·L^−1^ concentration. In concentrations of 1 and 10 g·L^−1^, an effect was reported after 1 and 10 min, respectively, but not with 0.25 g·L^−1^. Patin [51] also reported anesthetic effects of cocaine for animals in general. However, it has been withdrawn of the market because of the abuse potential and toxicity.

**Corrosive sublimate** (mercury (II) chloride or mercury bichloride, HgCl_2_) is a white crystalline solid and a laboratory reagent that is very toxic for humans. Its toxicity is due both to its mercury content and its corrosive properties. It was claimed to have anesthetic effects in crustaceans [35]. However, no anesthetic effects of this compound have been reported in any species so far, and it is likely this agent killed the animals rather than anesthetizing them. Since it is also highly toxic to people, its use is not recommended.

**Curare** is a common name for various plant extract alkaloids that have been used as neurotoxic poisoning for centuries by indigenous people in central and south America. Curare is a neuromuscular blocking agent that reversibly inhibits the nicotinic acetylcholine receptor at the neuromuscular junction. It was used to immobilize water fleas, including *E. coregoni*, *C. sphaericus*, *D. cucullata*, *D. brachyurum*, *E. graciloides*, and *M. leuckarti* [43]. However, since curare acts as a neuromuscular blocking agent, without effects on consciousness, it should not be used as an anesthetic.

**Methyl cellulose** (or methylcellulose) is a chemical compound derived from cellulose. Its anesthetic properties for small crustaceans were tested by immersion in a water bath with no success [70]. It is therefore not recommended as an anesthetic for any species of crustacean.

**Nicotine** is a naturally produced alkaloid that acts as a receptor agonist at most nicotinic acetylcholine receptors. Nicotine was assessed as an anesthetic in crayfish (*C. destructor*), but without success [65]. Although listed as an anesthetic for different animal species [51], its mechanisms of action suggest it should not be used for this purpose.

**Osmic acid**, also known as osmium tetroxide, is a high toxic chemical compound with the formula OsO_4_. It was used by Lo Bianco [35] to anesthetize *Evadne* and *Podon* species, as well as larvae of decapods, in a 1% concentration bath. However, its effects have been barely studied due to its high toxicity, and it is not an anesthetic agent, so it should not be used in for this purpose in any species.

**Propylene phenoxetol** acts as an anesthetic by inhibiting acetylcholine synthesis, inducing neural, muscular, and behavioral effects in gastropods [201]. Beside gastropods, it has been reported as an anesthetic for Amphipoda, Mysidacea, and Caridea species, relaxing their bodies with intact appendages when immersed in a bath for 15 min at a concentration of 0.15% [192]. This method is suggested to be appropriate for anesthetizing specimens before killing, especially for preservation.

**Tubocurarine**. Tubocurarine chloride is a neuromuscular blocking agent [202]. In *D. pulex*, exposure in a water bath immobilized animals in 180 min [44], and no effect was seen in *C. pagurus* and *C. maenas* [45]. Since its mode of action is by paralyzing the animals, not by producing anesthesia, it should not be used.

Thermic methods

**Emersion in air**. Transporting live shrimps, crayfishes, lobsters, and crabs in air was recommended by the Australian government in cool and moist conditions [12]. The temperature to which the air is cooled depends on the tolerances of the species. For instance, the Balmain bug (*Ibacus peronii*) should be stored at 6 °C while the tropical rock lobster (*Panulirus ornatus*) can be held at up to 25 °C. Some crabs should not be held out of water for extended periods (generally greater than 6 h), while the mud crab *Scylla serrata* can remain emersed for up to 3 days.

In an earlier study, Gardner [50] reported that the crab *P. gigas* was quiescent during the first 4–8 h emersed at 12 °C, while after 12 h it became active again. The quiescent state of the spanner crab *R. ranina* being stored in air was reported to be up to 3 days using low temperature and water sprays [190]. However, even in these conditions, emersed crabs presented the highest concentrations of lactate in the hemolymph compared to submersed crabs.

Hence, although decapods could seem to be unaffected by emersion due to their inactivity, they should not be held out of water for extended periods as the exchange of oxygen and carbon is curtailed and the animal rapidly asphyxiates, their gills tend to dry out, and general physiological disturbance occurs.

**Heating methods** have been used for narcotizing animals in order to preserve them in the best condition to be fixed for histological studies [51], but mainly to stun and humanely kill decapods for consumption [50,86,89,191,193]. More recently, Fregin and Bickmeyer used implanted electrodes to monitor signal propagation in the nervous system of lobsters and crayfishes to evaluate these methods [85].

Two heating methods have been proposed, placing crustaceans directly in hot or boiling water or a gradual increase in temperature for their preservation or stunning. Aversive reactions to hot water or boiling were reported for lobsters (*H. gammarus*) [191] and crabs (*C. pagurus*) [86,89], lasting between 2 and 10 min depending on species. Baker [89] reported that all legs were autotomized in 10 s when crabs were directly put in hot water. The crab *P. gigas*, which has an optimal temperature of 9–13 °C, was unaffected when placed at 17, 18, and 20 °C, but mild paralysis was observed at 24 °C in 2 h [50]. A study by Fregin and Bickmeyer [85] also indicated that movements and aversive behavior stopped in lobsters (*H. americanus*) and crayfishes (*A. astacus* and *A. leptodactylus*) when the animals were exposed to boiling water. However, they recorded a strong increase in electrophysiological signals after roughly 30 to 150 s, depending on animal size [85,86].

The first report concerning a gradual increase in temperature as the most humane killing method for lobsters was described in 1914 by Sinel in the European lobster (*H. gammarus*), who reported the unconscious state of lobsters at 18 °C and death at 26.6 °C [193]. Further studies in the edible crab *C. pagurus* showed loss of consciousness at 38–40 °C [86,89]. Lobsters and crayfishes lost consciousness at 30 °C and were killed when temperatures were slowly raised without any apparent signs of stress or movements [85]. Moreover, slowly rising water temperatures (1 °C min^−1^) did not cause unusual excitation of electrical activities to external stimuli in the central nervous system until reaching about 30 °C. Therefore, these authors suggested that slow warming is a usable method.

Nevertheless, the majority of the authors agree that heating methods appeared to cause distress to crustaceans, and the RSPCA in Australia [160] declared these methods unacceptable since they induce a degree of pain and suffering to the animals.

Osmotic methods

Immersion in tap water or **drowning** of marine crustaceans has been suggested by a few authors. An early study by Lo Bianco recommended immersion of decapods in freshwater to avoid breaking their appendages before transferring them into alcohol for preservation [35]. However, Baker demonstrate that the crab *C. pagurus* suffered 60% autotomy when placed in freshwater as well as increasing its activity and respiratory current, and the crabs exhibited strong uncoordinated movements [89]. Knudsen, later suggested the addition of 50 to 75% freshwater to kill marine crustaceans [70]. Gunter included a further step, heating the water to 40 °C, and reported that this treatment will kill the crabs quickly and easily without showing distress [194]. Drowning the crab *P. gigas* resulted in immediate rigidity and easy handling, however, after 10 min, the animals became very active, tore at their abdomens and walking legs, autotomy occurred, and they suffered mortality in less than 5 h [50], hence the authors reported that this method should not be used. The Australian RSPCA reported that placing live marine crustaceans in freshwater induces a severe osmotic shock which is likely to cause suffering prior to death and therefore they consider this method unacceptable [160].

### 3.3. Supportive Decision Tool

#### 3.3.1. Objective of the Tool

The objective of this publicly available on-line tool is to make information available regarding anesthetic methods for crustacean species. For the user’s convenience, this information can be consulted, filtered, and shared by species, group of species, anesthetic method, stage of development, sex, size, and environmental conditions used (temperature, salinity and light). We have included all the data that we could locate from published papers, reports, or books in the tool in order to help scientists to choose the best anesthetic method. Our goal is to provide an on-line tool that can be continuously updated with new information when this becomes available. We encourage readers to contact us if they have additional information that can improve included in the database.

#### 3.3.2. Explanation of the Decision Tool (How to Use It)

Some factors to consider before using the tool are the following: Time to induce anesthesia and recovery are related to full anesthesia, except whenever indicated in the comment column. We consider that anesthesia was reached when at least 50% of the population was anesthetized; when it is not 100% of the population that reach anesthesia, the percentage is indicated in the comment’s column. For the tool, the time unit has been homogenized to seconds, temperature in degree Celsius, salinity in ppt, light in lux, weight in grams, and measures in millimeters. For the concentration, three types of units are proposed that can be filtered; in addition, an extra column indicating the units cited in the publication is included. There is a column for the recommended concentration by the authors of the publication cited, and another column for the methods that the authors of this review consider unsuitable. All the data that were available for extraction from the publications are in the tool for easy access, with the reference of the publication (full citation and DOI whenever available), which will help the user to clarify the content of the study.

When a facet is selected, a filter is applied to the data. These are reflected in the boxes at the top of the page.

Sharing links. Every page within Datasette is designed to be shared using “copy and paste” of the page URL to share it with someone else. This includes applied filters and facets, thus, specific searches can be shared.Exporting data. The raw data can be exported. This is a fundamental principle of the project. There are CSV and .json links on each page to export the data in those formats.In the tool, there is a PDF tutorial with more detailed descriptions and examples of how to use it to facilitate readers to make the maximum use of the tool.

#### 3.3.3. Link of the Decision Tool

https://crusanest.icm.csic.es/, accessed on 4 December 2022.

## 4. Discussion

This review presents a comprehensive evaluation of drugs and methods used to provide analgesia and anesthesia in all crustaceans rather than just decapods [7]. This information is now in a searchable form that scientists and other users can employ to obtain relevant information on how to anesthetize their species of choice. Where this information is unavailable, since a relatively small number of species have been tested for each agent or technique, there is information on the drug, route of administration, dose, induction, and recovery times on closely related species since these data may be useful for pilot trials. The review and tool can, therefore, form the basis of choosing scientifically informed methods so that crustaceans can be rendered sedated or anesthetized during potentially painful or stressful procedures. Humane treatment of these animals improves their welfare and may mean that studies are not confounded by stress or pain [203]. De Sousa Valente [7] provides a useful overview of the techniques of anesthetizing decapods and proposes indicators of the different planes of levels of anesthesia. Whether these indicators can be used in non-decapod crustaceans remains to be seen, but they are a helpful important starting point. Future studies exploring analgesia or anesthesia in a species which has previously received no attention should consider ensuring all the facts surrounding the anesthesia technique are published, and then this new information can be added to the tool.

Several techniques and chemicals were, in our opinion, unsuitable for use as anesthetic in crustaceans. Many of the chemicals have no anesthetic properties and were used as fixatives in early studies. Given the evidence for the possibility of pain perception in decapods and that non-decapods have not been thoroughly studied when it comes to the existence of pain, we propose that researchers should treat the animals humanely and should apply the principles of good handling, care, and the management of stress and pain to safeguard the welfare of crustaceans. Techniques or chemicals that cause an adverse reaction should be avoided.

## 5. Conclusions

A range of methods and drugs are effective at providing anesthesia and analgesia in crustaceans. However, often for one technique or method, only a few species have been investigated; therefore, more research is needed in assessing a wider range of species. There are substantial differences among species and further age, developmental stage, sex, temperature, pH, and salinity may all influence the efficacy of a chosen method or drug. Therefore, the new tool provides an invaluable database that can be searched to find information on species, method, or chemical. This resource will allow researchers to make scientifically informed choices regarding anesthesia, but also identify gaps in our knowledge such that new avenues in research can be recognized and explored in future studies.

## Figures and Tables

**Table 1 biology-12-00387-t001:** List of analgesics used in crustaceans.

System	Subsystem	Class	Order	Species	References
Esters	Tricaine Methanesulphonate (MS 222) *	Malacostraca	Amphipoda	*Gammarus pulex*	[20]
Opioids	Morphine	Malacostraca	Decapoda	*Carcinus aestuarii*	[21]
				*Carcinus maenas*	[22]
				*Chasmagnathus granulatus*	[23,24,25]
				*Orconectes rusticus*	[26,27]
			Stomatopoda	*Squilla mantis*	[28]
Other organics	Lidocaine *	Malacostraca	Decapoda	*Macrobrachium americanum*	[29,30]
				*Litopennaeus vannamei*	

* used as local anesthetic.

**Table 2 biology-12-00387-t002:** List of anesthetic agents and other chemical used in crustaceans.

System	Subsystem	Class	Order	Species	References
Drugs acting at adrenoreceptors	Chlorpromazine hydrochloride	Malacostraca	Decapoda	*Cancer pagurus*	[45]
				*Carcinus maenas*	[45]
	Xylazine	Malacostraca	Decapoda	*Coenobita clypeatus*	[46]
				*Cancer pagurus*	[45]
				*Carcinus maenas*	[45]
Alcohols	2-phenoxy ethanol	Malacostraca	Decapoda	*Macrobrachiurn rosenbergii*	[47]
				*Sagmariasus verreauxi*	[48]
				*Eriocheir sinensis*	[49]
				*Pseudocarcinus gigas*	[50]
	Ethanol			Various species	[51]
		Branchiopoda freshwater species	Onychopoda	Branchiopoda freshwater species	[41]
		Branchiopoda	Anomopoda	*Daphnia magna*	[38,52,53]
			Anostraca	*Artemia franciscana*	[54]
			Onychopoda	Cladocera frehwater species	[41]
		Ostracoda		Ostracoda freshwater species	[41]
				Ostracoda species	[35]
		Hexanauplia (Subclasse Copepoda)	Cyclopodia	Copepoda freshwater species	[41]
				*Cyclops* sp.	[55]
		Thecostraca (Subclass Cirripedia)	Balanomorpha	*Balanus* spp.	[35]
			Scalpellomorpha	*Conchoderma* spp.	[35]
				*Lepas* spp.	[35]
		Malacostraca	Cumacea	Cumacea species	[35]
			Isopoda	Isopoda freshwater species	[41]
				Isopoda species	[35]
			Amphipoda	Amphipoda freshwater species	[41]
				*Gammarus minus*	[56]
			Schizopoda	Schizopoda species	[35]
			Decapoda	*Litopenaeus vanamei*	[57]
				*Litopenaeus vanamei*	[58]
				*Macrobrachium rosenbergii*	[59]
				*Macrobrachium tenellum*	[60]
				Crayfish species	[41]
				*Cherax destructor*	[61]
				*Nephrops norvegicus*	[62]
				*Cancer magister*	[63]
				*Hemigrapsus oregonensis*	[63]
				*Pugettia producta*	[63]
			Stomatopoda	Stomatopoda species	[35]
	Ether	Branchiopoda	Anomopoda	*Daphnia magna*	[38]
			Anostraca	*Artemia franciscana*	[64]
		Malacostraca	Decapoda	*Cherax destructor*	[65]
				*Homarus americanus*	[66]
				*Eriocheir sinensis*	[49]
	Isobutanol	Malacostraca	Decapoda	*Cherax destructor*	[65]
				*Homarus americanus*	[66,67]
				*Panulirus homarus*	[68]
				*Panulirus ornatus*	[68]
				*Panulirus polyphagus*	[68]
				*Panulirus versicolor*	[68]
				*Eriocheir sinensis*	[49]
			Stomatopoda	*Squilla mantis*	[69]
	Menthol			Various species	[51]
		Branchiopoda	Anomopoda	*Daphnia pulex*	[44]
		Hexanauplia (Subclasse Copepoda)	Calanoida	*Diaptomus* spp.	[44]
				*Limnocalanus macrurus*	[44]
			Cyclopodia	*Diacyclops bicuspidatus*	[44]
		Thecostraca (Subclass Cirripedia)		Cirripedia species	[42,70]
		Malacostraca	Decapoda	*Macrobrachium rosenbergii*	[71]
				*Macrobrachiurn tenellum*	[60]
				*Palaemonetes sinensis*	[72]
				*Cherax destructor*	[65]
				*Neohelice granulata*	[73]
	Methyl alcohol	Branchiopoda	Anomopoda	*Daphnia pulex*	[44]
		Hexanauplia (Subclasse Copepoda)	Calanoida	*Diaptomus* spp.	[44]
				*Limnocalanus macrurus*	[44]
			Cyclopodia	*Diacyclops bicuspidatus*	[44]
	Methyl pentynol	Branchiopoda	Anomopoda	*Daphnia pulex*	[44]
		Hexanauplia (Subclasse Copepoda)	Calanoida	*Diaptomus* spp.	[44]
				*Limnocalanus macrurus*	[44]
			Cyclopodia	*Diacyclops bicuspidatus*	[44]
		Malacostraca	Decapoda	*Homarus americanus*	[66]
	Tert-amyl alcohol	Branchiopoda	Anomopoda	*Daphnia pulex*	[44]
Esters	Benzocaine	Malacostraca	Decapoda	*Palaemon elegans*	[74]
				*Cherax destructor*	[65]
				*Cancer pagurus*	[45]
				*Carcinus maenas*	[45]
				*Pseudocarcinus gigas*	[50]
	Pantocaine	Branchiopoda	Anomopoda	*Eubosmina coregoni*	[43]
				*Chydorus sphaericus*	[43]
				*Daphnia cucullata*	[43]
			Ctenopoda	*Diaphanosoma brachyurum*	[43]
		Hexanauplia (Subclasse Copepoda)	Calanoida	*Eudiaptomus graciloides*	[43]
			Cyclopodia	*Mesocyclops leuckarti*	[43]
	Phenyluretane	Branchiopoda	Anomopoda	*Daphnia magna*	[38]
	Procaine	Branchiopoda	Anomopoda	*Daphnia magna*	[38]
		Malacostraca	Decapoda	*Cancer pagurus*	[45]
				*Carcinus maenas*	[45]
	Tetracaine hydrochloride	Branchiopoda	Anomopoda	*Daphnia magna*	[75]
	Tricaine Methane Sulfate (MS 222)	Branchiopoda	Anostraca	*Artemia franciscana*	[54]
			Anomopoda	*Daphnia pulex*	[44]
		Hexanauplia (Subclasse Copepoda)	Calanoida	*Diaptomus* spp.	[44]
				*Limnocalanus macrurus*	[44]
			Cyclopodia	*Diacyclops bicuspidatus*	[44]
		Ostracoda	Podocopida	*Eucypris virens*	[76]
		Malacostraca	Amphipoda	*Corophium volutator*	[77]
				*Echinogammarus obtusatus*	[77]
				*Gammarus pulex*	[20]
			Decapoda	*Macrobrachium rosenbergii*	[78]
				*Astacus astacus*	[79]
				*Orconectes virilis*	[80]
				*Homarus americanus*	[66]
				*Cancer magister*	[63]
				*Cancer pagurus*	[45]
				*Carcinus maenas*	[45]
				*Crangon septemspinosa*	[81]
				*Eriocheir sinensis*	[49]
				*Hemigrapsus nudus*	[82]
				*Hemigrapsus oregonensis*	[63]
				*Petrolisthes cinctipes*	[82]
				*Pseudocarcinus gigas*	[50]
				*Pugettia producta*	[63]
	Urethane			Various species	[51]
		Branchiopoda	Anomopoda	*Daphnia magna*	[38]
				*Daphnia pulex*	[44]
				*Daphnia* spp.	[37,40]
		Hexanauplia (Subclasse Copepoda)	Calanoida	*Diaptomus* spp.	[44]
				*Limnocalanus macrurus*	[44]
			Cyclopodia	*Cyclops* sp.	[37]
				*Diacyclops bicuspidatus*	[44]
		Malacostraca	Isopoda	*Asellus* spp.	[37]
				*Idotea* spp.	[37]
			Decapoda	*Astacus fluviatilis*	[37]
				*Astacus* sp.	[40]
Steroid	Alfaxalone	Malacostraca	Decapoda	*Callinectes sapidus*	[83]
				*Cancer pagurus*	[45]
				*Carcinus maenas*	[45]
Inhalant (Gaseous)	Carbon Dioxide			Various species	[51]
		Branchiopoda	Anomopoda	*Daphnia pulex*	[44]
		Hexanauplia (Subclasse Copepoda)	Calanoida	*Diaptomus* spp.	[44]
				*Limnocalanus macrurus*	[44]
			Cyclopodia	*Diacyclops bicuspidatus*	[44]
		Thecostraca (Subclass Cirripedia)	Balanomorpha	*Balanus* spp.	[84]
		Malacostraca	Decapoda	*Palaemon longirostris*	[84]
				*Cangron cangron*	[84]
				*Astacus astacus*	[85]
				*Astacus leptodactylus*	[85]
				*Cherax destructor*	[65]
				*Homarus americanus*	[85]
				*Homarus gammarus*	[85]
				*Cancer pagurus*	[86]
				*Portunus sanguinolentus*	[87]
				*Pseudocarcinus gigas*	[50]
	Cycloprane	Branchiopoda	Anostraca	*Artemia franciscana*	[64]
	Nitrogen	Malacostraca	Decapoda	*Cancer pagurus*	[86]
				*Macrobrachium rosenbergii*	[88]
Inhalant (Volatile)	Chloroform	Branchiopoda	Anomopoda	*Daphnia pulex*	[44]
			Anostraca	*Artemia franciscana*	[64]
		Hexanauplia (Subclasse Copepoda)	Calanoida	*Diaptomus* spp./ *Limnocalanus macrurus*	[44]
			Cyclopodia	*Diacyclops bicuspidatus*	[44]
		Malacostraca	Decapoda	*Astacus* spp.	[42]
				*Cancer pagurus*	[89]
				*Carcinus* spp.	[42]
				*Cherax destructor*	[65]
				*Eriocheir sinensis*	[49]
				*Homarus americanus*	[66]
				*Homarus* spp.	[42]
				*Pseudocarcinus gigas*	[50]
	Enflurane	Branchiopoda	Anomopoda	*Daphnia magna*	[52,90]
	Halotane	Branchiopoda	Anomopoda	*Daphnia magna*	[52,90]
			Anostraca	*Artemia franciscana*	[64]
		Malacostraca	Decapoda	*Astacus astacus*	[79]
				*Cherax destructor*	[65]
				*Litopenaeus vannamei*	[91]
	Isoflurane	Branchiopoda	Anomopoda	*Daphnia magna*	[90]
		Malacostraca	Decapoda	*Eriocheir sinensis*	[49]
Injectables	Bupivacaine hydrochloride monohydrate	Branchiopoda	Anomopoda	*Daphnia magna*	[75]
	Chloral hydrate			Varoius species	[51]
		Branchiopoda	Anomopoda	*Eubosmina coregoni*	[43]
				*Chydorus sphaericus*	[43]
				*Daphnia cucullata*	[43]
				*Daphnia magna*	[38]
				*Daphnia pulex*	[44]
				*Moina macrocopa*	[41]
		Branchiopoda	Ctenopoda	*Diaphanosoma brachyurum*	[43]
				*Sididae species*	[41]
		Hexanauplia (Subclasse Copepoda)	Calanoida	*Diaptomus* spp.	[44]
				*Eudiaptomus graciloides*	[43]
				*Limnocalanus macrurus*	[44]
			Cyclopodia	*Diacyclops bicuspidatus*	[44]
				*Mesocyclops leuckarti*	[43]
	Etomidate	Malacostraca	Decapoda	*Callinectes sapidus*	[92]
				*Eriocheir sinensis*	[49]
	Ketamine	Malacostraca	Decapoda	*Callinectes sapidus*	[92]
				*Orconectes virilis*	[80]
				*Pseudocarcinus gigas*	[50]
	Pentobarbital	Branchiopoda	Anomopoda	*Daphnia pulex*	[44]
		Malacostraca	Decapoda	*Callinectes sapidus*	[92]
				*Cancer pagurus*	[45]
				*Carcinus maenas*	[45]
	Propanidid	Malacostraca	Decapoda	*Cancer pagurus*	[45]
				*Carcinus maenas*	[45]
	Propofol	Malacostraca	Decapoda	*Callinectes sapidus*	[92]
	Quinaldine	Branchiopoda	Anomopoda	*Daphnia pulex*	[44]
		Hexanauplia (Subclasse Copepoda)	Calanoida	*Diaptomus* spp.	[44]
				*Limnocalanus macrurus*	[44]
			Cyclopodia	*Diacyclops bicuspidatus*	[44]
		Malacostraca	Decapoda	*Homarus americanus*	[66]
				*Macrobrachium rosenbergii*	[78]
	Tiletamine–zolazepam	Malacostraca	Decapoda	*Callinectes sapidus*	[92]
Oils	Aqui-STM	Malacostraca	Decapoda	*Cancer pagurus*	[93]
				*Jasus edwardsii*	[94]
				*Macrobrachium rosenbergii*	[78]
				*Pseudocarcinus gigas*	[50]
	Clove oil (CO)	Branchiopoda	Anomopoda	*Daphnia magna*	[95,96]
		Malacostraca	Amphipoda	*Gammarus acherondytes*	[56]
				*Gammarus minus*	[56]
			Decapoda	*Cancer magister*	[63]
				*Cherax destructor*	[65]
				*Cherax quadricarinatus*	[97]
				*Coenobita clypeatus*	[98]
				*Eriocheir sinensis*	[49]
				*Hemigrapsus oregonensis*	[63]
				*Macrobrachium rosenbergii*	[78,59]
				*Macrobrachium tenellum*	[60]
				*Penaeus brasiliensis*	[99]
				*Penaeus schmitti*	[99]
				*Penaeus semisulcatus*	[100]
				*Portunus sanguinolentus*	[87]
				*Pseudocarcinus gigas*	[50]
				*Pugettia producta*	[63]
				*Sagmariasus verreauxi*	[48]
	Essential oils of *Lippia alba* (EOLA)	Malacostraca	Decapoda	*Penaeus paulensis*	[58]
				*Litopenaeus vanamei*	[57]
				*Neohelice granulata*	[73]
	Essential oils of *Aloysia triphylla* (EOAT)	Malacostraca	Decapoda	*Litopenaeus vannamei*	[57]
				*Neohelice granulata*	[73]
	Essential oils of *Cymbopogon citratus* (EOC)	Malacostraca	Decapoda	*Litopenaeus vannamei*	[101]
	Essential oils of *Ocimum gratissimum* (EOOG)	Malacostraca	Decapoda	*Penaeus paulensis*	[58]
	Essential oils of *Origanum majorana* (EOO)	Malacostraca	Decapoda	*Litoenaeus vannamei*	[58]
	Essential oils of *Lavandula officinalis* (LAV)	Malacostraca	Decapoda	*Penaeus brasiliensis*	[99]
				*Penaeus schmitti*	[99]
	Essential oils of *Amyris balsamifera* (SAN)	Malacostraca	Decapoda	*Penaeus brasiliensis*	[99]
				*Penaeus schmitti*	[99]
	Essential oils of *Mentha piperita* (MEN)	Malacostraca	Decapoda	*Penaeus brasiliensis*	[99]
				*Penaeus schmitti*	[99]
	Essential oils of *Melaleuca alternifolia* (tea tree oil: TTO)	Malacostraca	Decapoda	*Neohelice granulata*	[73]
	Essential oils of *Passiflora incarnata*	Malacostraca	Decapoda	*Macrobrachium tenellum*	[60]
	Essential oils of *Valeriana officinalis*	Malacostraca	Decapoda	*Macrobrachium tenellum*	[60]
	Eugenol	Branchiopoda	Anostraca	*Artemia franciscana*	[54]
		Malacostraca	Amphipoda	*Gammarus pulex*	[20]
			Decapoda	*Penaeus chiniensis*	[102]
				*Penaeus indicus*	[103]
				*Penaeus japonicus*	[104,105]
				*Penaeus monodon*	[106]
				*Litopenaeus vannamei*	[57,107]
				*Macrobrachium rosenbergii*	[71,88]
				*Palaemonetes sinensis*	[108]
				*Astacus astacus*	[107]
				*Cherax destructor*	[65]
				*Procambarus clarkii*	[109]
				*Homarus americanus*	[110,111]
				*Nephrops norvegicus*	[62]
				*Callinectes sapidus*	[107]
				*Cancer irroratus*	[110]
				*Carcinus maenas*	[110]
				*Neohelice granulata*	[73]
Other organic compounds	Acetic paraldehyde	Branchiopoda	Anomopoda	*Daphnia longispina*	[39]
				*Daphnia obtusa*	[39]
		Hexanauplia (Subclasse Copepoda)	Cyclopodia	*Cyclops strenuus*	[39]
				*Eudiaptomus vulgaris*	[39]
				Copepoda species	[36]
	Chlorotone			Various species	[51]
		Branchiopoda	Anomopoda	*Daphnia pulex*	[44]
				*Daphnia* spp.	[36]
		Cirripedia		Cirripidia species	[70]
		Hexanauplia (Subclasse Copepoda)	Calanoida	*Diaptomus* spp.	[44]
				*Limnocalanus macrurus*	[44]
			Cyclopodia	*Diacyclops bicuspidatus*	[44]
		Ostracoda		Ostracoda species	[36]
	Clomethiazole (Hemineurine)	Malacostraca	Decapoda	*Carcinus maenas*	[112]
				*Necora puber*	[112]
	Guaiacol glyceryl ether	Malacostraca	Decapoda	*Cancer pagurus*	[45]
				*Carcinus maenas*	[45]
	Lidocaine	Branchiopoda	Anomopoda	*Daphnia magna*	[75]
		Malacostraca	Decapoda	*Callinectes sapidus*	[92]
				*Cancer pagurus*	[45]
				*Carcinus maenas*	[45]
				*Macrobrachium americanum*	[30]
				*Orconectes virilis*	[80]
				*Litopenaeus vannamei*	[29,91]
				*Portunus sanguinolentus*	[87]
				*Pseudocarcinus gigas*	[50]
			Stomatopoda	*Squilla mantis*	[69]
	Piperazinum	Branchiopoda	Anomopoda	*Eubosmina coregoni*	[43]
				*Chydorus sphaericus*	[43]
				*Daphnia cucullata*	[43]
			Ctenopoda	*Diaphanosoma brachyurum*	[43]
		Hexanauplia (Subclasse Copepoda)	Calanoida	*Eudiaptomus graciloides*	[43]
			Cyclopodia	*Mesocyclops leuckarti*	[43]
	Suxamethonium chloride	Malacostraca	Decapoda	*Cancer pagurus*	[45]
				*Carcinus maenas*	[45]
	Terpinen-4-ol	Malacostraca	Decapoda	*Neohelice granulata*	[73]
				Various species	[51]

**Table 3 biology-12-00387-t003:** List of physical methods used in crustaceans.

System	Subsystem	Class	Order	Species	References
Electrical	Electric shock (stunning)	Malacostraca	Decapoda	*Astacus astacus*	[85]
				*Astacus leptodactylus*	[85]
				*Callinectes sapidus*	[152]
				*Cancer pagurus*	[86,153,154,155,156]
				*Carcinus maenas*	[157]
				*Homarus americanus*	[85]
				*Hommarus gammarus*	[153,154]
				*Nephrops norvegicus*	[157,158]
				*Litopenaeus vannamei*	[152]
				*Procambarus clarkii*	[152,159]
				Decapoda species	[160]
Mechanical	Pithing	Malacostraca	Decapoda	Bracyura species	[160]
				*Cherax destructor*	[12]
				*Cherax quadricarinatus*	[12]
				*Cherax tenuimanus*	[12]
				Decapods but brachyura	[160]
				*Ibacus peronii*	[12]
				*Jasus edwardsii*	[12]
				*Panulirus cygnus*	[12]
				*Panulirus ornatus*	[12]
				*Thenus orientalis*	[12]
	Spiking	Malacostraca	Decapoda	*Cancer pagurus*	[86,89,161]
				*Scylla serrata*	[12]
Thermic	Chilling in ice/water slurry	Hexanauplia (Subclasse Copepoda)	Cyclopodia	*Cyclops* sp.	[55]
		Malacostraca	Decapoda	*Astacus astacus*	[85]
				*Astacus leptodactylus*	[85]
				*Callinectes sapidus*	[152]
				*Cancer pagurus*	[86,157,161]
				*Carcinus maenas*	
				*Cherax destructor*	[12,65]
				*Cherax quadricarinatus*	[12]
				*Cherax tenuimanus*	
				*Eriocheir sinensis*	[49]
				*Fenneropenaeus merguiensis*	[12]
				*Homarus americanus*	[85,157]
				*Ibacus peronii*	[12]
				*Jasus edwardsii*	[12]
				*Maja brachydactyla*	[162]
				*Metapenaeus ensis*	[12]
				*Nephrops norvegicus*	[163]
				*Orconectes rusticus*	[164]
				*Panulirus cygnus*	[12]
				*Panulirus ornatus*	[12]
				*Penaeus esculentus*	[12]
				*Penaeus japonicus*	[12]
				*Penaeus monodon*	[12,182]
				*Litopenaeus vannamei*	[152]
				*Portunus pelagicusus*	[165]
				*Portunus sanguinolentus*	[87]
				*Procambarus clarkii*	[152,166,167]
				*Pseudocarcinus gigas*	[50]
				*Scylla serrata*	[12]
				*Thenus orientalis*	[12]
				Tropical and temperate species	[160]
				Marine tropical crabs	[70]
	Chilling in water	Malacostraca	Decapoda	*Cherax tenuimanus*	[12]
				*Macrobrachium rosenbergii*	[88,168,169]
				*Pacifastacus leniusculus trowbridgii*	[136]
				*Penaeus monodon*	[169]
				*Cherax tenuimanus*	[12]
				*Penaeus chiniensis*	[170]
				*Penaeus esculentus*	[171]
				*Penaeus japonicus*	[172,173,174,175,176,177]
				*Penaeus monodon*	[12,169,171,178]
				*Penaeus semisulcatus*	[171]
				*Litopenaeus vannamei*	[179]
	Dry chilling	Malacostraca	Decapoda	*Cancer pagurus*	[86]
				*Cherax destructor*	[12]
				*Cherax quadricarinatus*	[12]
				*Cherax tenuimanus*	[12]
				Crab species	[169]
				*Fenneropenaeus merguiensis*	[12]
				*Homarus americanus*	[169]
				*Ibacus peronii*	[12]
				*Jasus edwardsii*	[12]
				Large crustaceans adapted to very low temperatures	[160]
				*Lithodes santolla*	[180]
				*Macrobrachium rosenbergii*	[88,168]
				*Metapenaeus ensis*	[12]
				*Orconectes rusticus*	[26]
				*Panulirus cygnus*	[12]
				*Panulirus ornatus*	[12]
				*Penaeus chiniensis*	[170,171]
				*Penaeus esculentus*	[12]
				*Penaeus indicus*	[176]
				*Penaeus japonicus*	[169,174,175][12,171,172,173,176]
				*Penaeus monodon*	[12,178]
				*Penaeus semisulcatus*	[171]
				*Litopenaeus vannamei*	[179]
				*Scylla serrata*	[12]
				*Thenus orientalis*	[12]
				Warm water lobster species	[169]

**Table 4 biology-12-00387-t004:** Methods suggested as unsuitable for use as anesthetics in crustaceans.

Method	System	Subsystem	Class	Order	Species	References
Chemical	Alcohol	BOUIN (Alcohol + Formaldehyde + glacial acetic acid + picric acid)			Various species	[51]
			Branchiopoda	Anostraca	Anostraca species	[42]
			Ostracoda		Ostracoda species	[42]
			Malacostraca	Decapoda	*Astacus* spp.	[42]
					*Homarus* spp.	[42]
					*Carcinus* spp.	[42]
	Other organic compounds	Butyn	Branchiopoda	Anomopoda	*Daphnia pulex*	[44]
			Hexanauplia (Subclasse Copepoda)	Calanoida	*Diaptomus* spp.	[44]
					*Limnocalanus macrurus*	[44]
				Cyclopodia	*Diacyclops bicuspidatus*	[44]
		Decamethonium bromide	Malacostraca	Decapoda	*Cancer pagurus*	[45]
					*Carcinus maenas*	[45]
		Gallamine Triethiodite	Malacostraca	Decapoda	*Cancer pagurus*	[45]
					*Carcinus maenas*	[45]
		Guaiacol glyceryl ether	Malacostraca	Decapoda	*Cancer pagurus*	[45]
					*Carcinus maenas*	[45]
		Formalin			Crustacean species	[187]
			Branchiopoda		Branchiopoda species	[41]
			Branchipoda	Anostraca	Anostraca species	[187]
				Onychopoda (Cladocera)	Anostraca species	[187]
				Anostraca	Branchipoda species	[42]
			Ostracoda		Ostracoda species	[42]
			Hexanauplia (Subclasse Copepoda)		Copepoda species	[42]
			Malacostraca	Amphipoda	Amphipoda species	[187]
				Cumacea	Cumacea species	[187]
			Malacostraca	Decapoda and Stomatopoda	Malacostraca species	[187]
		Hydrogen peroxide	Branchiopoda	Anomopoda	*Daphnia pulex*	[44]
			Hexanauplia (Subclasse Copepoda)	Calanoida	*Diaptomus* spp.	[44]
					*Limnocalanus macrurus*	[44]
				Cyclopodia	*Diacyclops bicuspidatus*	[44]
		Hydroxylamina	Branchiopoda	Anomopoda	*Daphnia pulex*	[44]
			Hexanauplia (Subclasse Copepoda)	Calanoida	*Diaptomus* spp.	[44]
					*Limnocalanus macrurus*	[44]
				Cyclopodia	*Diacyclops bicuspidatus*	[44]
		Methysergide maleate	Malacostraca	Decapoda	*Procambarus clarkii*	[188]
	Salts	Magnesium chloride			Various species	[51]
			Malacostraca	Decapoda	*Astacus astacus*	[85]
					*Astacus leptodactylus*	[85]
					*Cherax destructor*	[65]
					*Homarus americanus*	[85]
		Magnesium sulphate	Branchiopoda	Anomopoda	*Daphnia pulex*	[44]
			Malacostraca	Decapoda	*Hemigrapsus sanguineus*	[2]
					*Homarus americanus*	[66]
					*Portunus sanguinolentus*	[87]
					*Pseudocarcinus gigas*	[50]
		Physostigmine salicylate	Branchiopoda	Anomopoda	*Eubosmina coregoni*	[43]
					*Chydorus sphaericus*	[43]
					*Daphnia cucullata*	[43]
					*Daphnia pulex*	[44]
				Ctenopoda	*Diaphanosoma brachyurum*	[43]
			Hexanauplia (Subclasse Copepoda)	Calanoida	*Diaptomus* spp.	[44]
					*Eudiaptomus graciloides*	[43]
					*Limnocalanus macrurus*	[44]
				Cyclopodia	*Diacyclops bicuspidatus*	[44]
					*Mesocyclops leuckarti*	[43]
		Potassium chloride	Malacostraca	Decapoda	*Cancer pagurus*	[86]
					*Homarus americanus*	[189]
		Potassium hydroxide	Chthyostraca (Subclass Branchiura)	Decapoda	Branchiura species	[42]
		Sodium bicarbonate	Malacostraca	Decapoda	*Ranina ranina*	[190]
		Sodium chloride	Malacostraca	Decapoda	*Cancer pagurus*	[86,89]
					*Homarus gammarus*	[191]
					*Pseudocarcinus gigas*	[50]
	Others	Albumin	Branchiopoda	Anomopoda	*Daphnia magna*	[38]
		Corrosive sublimate	Branchiopoda	Onychopoda	*Evadne* spp.	[35]
					*Podon* spp.	[35]
			Hexanauplia (Subclasse Copepoda)		Copepoda species	[35]
			Cumacea		Cumacea species	[35]
			Malacostraca	Amphipoda	*Phronima* spp.	[35]
				Decapoda	Decapoda species	[35]
				Isopoda	*Bopyroides* spp.	[35]
					*Entoniscoides* spp.	[35]
				Schizopoda	Schizopoda species	[35]
				Stomatopoda	Stomatopoda species	[35]
		Curare	Branchiopoda	Anomopoda	*Eubosmina coregoni*	[43]
					*Chydorus sphaericus*	[43]
					*Daphnia cucullata*	[43]
				Ctenopoda	*Diaphanosoma brachyurum*	[43]
			Hexanauplia (Subclasse Copepoda)	Calanoida	*Eudiaptomus graciloides*	[43]
				Cyclopodia	*Mesocyclops leuckarti*	[43]
		Osmic acid	Branchiopoda	Onychopoda	*Evadne* spp.	[35]
					*Podon* spp.	[35]
			Malacostraca	Decapoda	Decapoda species	[35]
		Physostigmine salicylate	Branchiopoda	Anomopoda	*Eubosmina coregoni*	[43]
					*Chydorus sphaericus*	[43]
					*Daphnia cucullata*	[43]
				Ctenopoda	*Diaphanosoma brachyurum*	[43]
		Propylene phenoxetol	Malacostraca	Amphipoda	Amphipoda species	[192]
				Decapoda	Caridea species	[192]
				Mysidacea	Mysidacea species	[192]
		Tubocurarine	Branchiopoda	Anomopoda	*Daphnia pulex*	[44]
			Malacostraca	Decapoda	*Cancer pagurus*	[45]
					*Carcinus maenas*	[45]
		Methyl cellulose			Small crustacean species	[70]
		Nicotine	Malacostraca	Decapoda	*Cherax destructor*	[65]
Physical	Emersion in air		Malacostraca	Decapoda	*Cherax destructor*	[12]
					*Cherax tenuimanus*	[12]
					*Fenneropenaeus merguiensis*	[12]
					*Ibacus peronii*	[12]
					*Jasus edwardsii*	[12]
					*Metapenaeus ensis*	[12]
					*Panulirus cygnus*	[12]
					*Panulirus ornatus*	[12]
					*Penaeus esculentus*	[12]
					*Penaeus japonicus*	[12]
					*Penaeus monodon*	[12]
					*Pseudocarcinus gigas*	[50]
					*Ranina ranina*	[190]
					*Scylla serrata*	[12]
					*Thenus orientalis*	[12]
	Heating		Crustaceans		Various species	[51]
			Malacostraca	Decapoda	Various species	[160]
			Malacostraca	Decapoda	*Astacus astacus*	[85]
					*Astacus leptodactylus*	[85]
					*Cancer pagurus*	[86,89]
					*Homarus americanus*	[85]
					*Homarus gammarus*	[191,193]
					*Pseudocarcinus gigas*	[50]
	Osmotic (immersion in tap water)		Malacostraca	Decapoda	Marine large crustaceans	[187]
					Decapoda species	[70]
						[160]
						[35]
					Crab and lobster species	[194]
					*Cancer pagurus*	[89]
					*Pseudocarcinus gigas*	[50]

## Data Availability

Data is contained within the article. The data presented in this study are available in https://crusanest.icm.csic.es/ (accessed on 2 February 2023).

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
