# Peer review of "Methods to Induce Analgesia and Anesthesia in Crustaceans: A Supportive Decision Tool"

_biology, 2023, doi:10.3390/biology12030387_

Round 1

Author Response

We are grateful to the reviewers for their time and useful comments that have greatly improved our manuscript. We thank the editor for the opportunity to resubmit this improved version.

Reviewer 1

Analgesia and anesthesia in crustaceans

This manuscript aims to provide a comprehensive overview of analgesics and anesthetics that have been used on crustaceans. It appears to be thorough and for most of the substances a mode of action is indicated. For anyone wishing to conduct research that requires protection against possible pain in these animals this will be a great resource. Further, the authors provide a link to a decision-making tool that will help in getting the appropriate treatment. Thus, I firmly support the aims of this work.

It is a long manuscript, and the nature of the content does not make for an exciting read. It is inclined to list potential substances, in a logical order, and to indicate what is known for each. I suspect the structure cannot be changed much, if at all. However, the major problem is the very poor standard of English that makes reading very difficult. I am surprised it was submitted in the present state. The manuscript requires a very though, line by line, editing to get this right. There are errors in spelling, redundant words and phrases, incorrect words and phrases and simply labored writing. These are so extensive that it is too much to ask a reviewer to correct the work.

Errors, typos and grammar have been corrected. Native English-speakers have revised the English style to make the MS fluently readable. Moreover, we included an index (table of contents) in the MS, to facilitate reader to go to specific sections, as suggested by reviewer 2.

I have some comments and queries about the use of specific terms and words, and I list these by line number.

51 Formalin is said to be a pain-causing chemical, but it was used to investigate if the animal showed responses that might indicate pain. Thus, it cannot be called pain-causing in this context. It might be clarified to say pain-causing in vertebrates.

It has been included that formalin is a pain-causing chemical in vertebrates.

54 Delete “(no pain)

(no pain) has been deleted.

55 use “a painful effect”

“the painful effect” has been changed to “a painful effect”.

59 were not seen

“was” has been changed to “were”.

60 avoid stating categorically that the claws were painfully treated

Sentence has been re-phrased.

80 change “pain” to “noxious stimuli”

“pain” has been changed to “noxious stimuli”.

86 We need to link to the reference for this study.

A reference has been cited.

135 Pain is not transmitted.

The reviewer is correct, what is transmitted are the neural signals not pain. Text has been modified.

143-4 This needs some work. I do not think that a passing shadow was used to induce pain. Please explain striking and switching.

This paragraph has been improved to better understand the effect of morphine in crustaceans as an analgesic.

159 What is flicked? How does a decrease in rubbing, non-sheltering and recoil indicate pain? It would help to know more about these responses. What is flicked? If recoil is a rapid move away it might indicate reflex response. The meaning is not clear because I am not sure whether the term indicator of pain means that the reduction in response is due to the substance and that might suggest pain would have been felt without the substance.

Flicking refers to tail flicking. Diarte-Plata et al. (2012) states that when applying lidocaine to the eyestalk area of prawns before ligation, behaviors such as tail flicking, disorientation, recoil, stooping, rubbing and non-sheltering were reduced. These behaviors are directly related to potential pain and discomfort since prawns where eyestalks are ablated without lidocaine displayed these behavioral responses. Sentence has been modified to clarify this meaning.

195 explain antihetic. It is not a term I know.

It is a spelling mistake; it should have said anesthetic.

238 delete “experimental”

‘Experimental’ has been deleted.

337 We do not need to know that these are from a particular lake

Name of the lake has been deleted.

341 What are water flies?

We were meaning water fleas, daphnids. It was a spelling mistake, it has been corrected.

377 Human or humane?

We were meaning human; it has been corrected.

399-406 Is Tricane Methane Sulfate the same as MS 222? If so, put the latter in brackets after the title

Yes, it is MS 222. Added.

440 Delete perception.

“Perception” has been deleted.

452 Give the name of the author

Author name has been added. The reference corresponds to Smaldon (1978).

494 some commercial species are small.

The reviewer is correct, we wanted to highlight that the effect of chloroform is related to the size of the animals. The sentence has been clarified.

542 Some of the uses of injectables did not use injection. Further, some other substances not described (mentioned in later stages) as injectables were injected. This is confusing.

We have classified alfaxalone as a steroid since we have given preference in the classification to the chemical nature. The reviewer is correct that not all the chemicals included in the injectables section have not been used as injectables in crustaceans but we believe that this is the most consistent classification for these compounds as it is the way they are commonly referred to in vertebrates.

629 What species?

The mode of action has been described in vertebrates, including fish. This information has been added to the MS.

I support the aims of the manuscript, but it needs considerable editing to make it readable

Editing and English style has been revised.

Reviewer 2 Report

The manuscript entitled “Methods to induce analgesia and anesthesia in crustaceans: a supportive decision tool” summarized the methods for anaesthetizing different crustacean species, and established an online database. This work provides a valuable dataset for several areas of crustacean studies. In general, the paper is clearly written and well organized. I think this paper is acceptable for publication in Biology. I just have a couple of suggestions:

1. as the authors states that there are grey literatures, and there are some papers in the languages beside English, I think the authors, if possible, could collaborate with the scientists in different countries to collect the information the authors can’t get. However, if the authors have difficulty in doing it, I would suggest them to extend their online database by adding the information provided by the scientists in different countries.

2. the paper is quite long, I would suggest to add a catalog at the beginning of the paper, then the readers can find the contents they are interested in easily.

3. there are some typos and the rules for using bold and Italian fonts seems inconsistent. The authors need to check it carefully.    

Author Response

Reviewer 2

The manuscript entitled “Methods to induce analgesia and anesthesia in crustaceans: a supportive decision tool” summarized the methods for anaesthetizing different crustacean species, and established an online database. This work provides a valuable dataset for several areas of crustacean studies. In general, the paper is clearly written and well organized. I think this paper is acceptable for publication in Biology. I just have a couple of suggestions:

  1. As the authors states that there are grey literatures, and there are some papers in the languages beside English, I think the authors, if possible, could collaborate with the scientists in different countries to collect the information the authors can’t get. However, if the authors have difficulty in doing it, I would suggest them to extend their online database by adding the information provided by the scientists in different countries.

The reviewer is correct; ­­ to our knowledge we have included all information about anesthetic methods in crustaceans, however we could miss some publications or reports of interest. Readers can contact us and provide additional information to keep improving the online tool. This information has been included in the MS.

  1. The paper is quite long, I would suggest to add a catalog at the beginning of the paper, then the readers can find the contents they are interested in easily.

An index (table of contents) has been included in the MS to facilitate the reader to go to a specific section.

  1. there are some typos and the rules for using bold and Italian fonts seems inconsistent. The authors need to check it carefully.

Errors, typos and wording has been corrected and English improved and edited.

Round 2

Reviewer 1 Report

The authors have attended to the specific points I made in the first review. However, the major problem noted then is that the style of writing rendered the manuscript virtually unreadable. The authors state in their response that they have attended to this, however, I see no evidence of such work. The writing remains very poor. Just one example is on lines 343-345 in which the word anesthesia appears three times, but with different spelling for each.  On this second review I started to highlight problems so as to provide more guidance about what is required and that is attached. However, the time taken for this was more than can be reasonably asked of a reviewer, especially as the authors had shown no enthusiasm for correction. I stopped after several hours at page 27 as I have other pressing matters. Should the authors wish to correct the manuscript they should note that the lack of highlights and suggestions after page 27 does NOT mean that I find it acceptable. Rather it means that I gave up reading. 

It is a great pity that the authors have left the manuscript in such a sorry state because I support the idea of the review. It could have been a good contribution but without effort from the authors to make it readable it remains unpublishable.

Author Response

All  comments reviewer 1 included in the manuscript as errors, typos and grammar have been corrected by the co-authors. Then professor, Flecknell, native English-speaker, has realized a complete editing of the manuscript to make the manuscript fluently readable.
